# Gene regulation gravitates toward either addition or multiplication when combining the effects of two signals

Eric M Sanford[1], Benjamin L Emert[1], Allison Coté[2,3], Arjun Raj[2,3]*

[1]Genomics and Computational Biology Graduate Group, Perelman School of Medicine, University of Pennsylvania, Philadelphia, United States; [2]Department of Bioengineering, School of Engineering and Applied Sciences, University of Pennsylvania, Philadelphia, United States; [3]Department of Genetics, Perelman School of Medicine, University of Pennsylvania, Philadelphia, United States

**Abstract** Two different cell signals often affect transcription of the same gene. In such cases, it is natural to ask how the combined transcriptional response compares to the individual responses. The most commonly used mechanistic models predict additive or multiplicative combined responses, but a systematic genome-wide evaluation of these predictions is not available. Here, we analyzed the transcriptional response of human MCF-7 cells to retinoic acid and TGF-β, applied individually and in combination. The combined transcriptional responses of induced genes exhibited a range of behaviors, but clearly favored both additive and multiplicative outcomes. We performed paired chromatin accessibility measurements and found that increases in accessibility were largely additive. There was some association between super-additivity of accessibility and multiplicative or super-multiplicative combined transcriptional responses, while sub-additivity of accessibility associated with additive transcriptional responses. Our findings suggest that mechanistic models of combined transcriptional regulation must be able to reproduce a range of behaviors.

*For correspondence:
arjunraj@seas.upenn.edu

## Introduction

Suppose a cell at baseline expresses 100 copies of mRNA of gene X. If you give signal A, the cell expresses 200 copies of gene X. Give signal B, and you see 300 copies. What happens when you give both signals at the same time? Do the effects add (gene X increases to 400 copies)? Multiply (600 copies)? Additive and to some extent multiplicative phenomenological models have seen widespread use due to their simple mechanistic basis. However, there is little systematic empirical evidence that either of these phenomenological models of combined responses are in general valid or should be favored in any way.

Part of the appeal of the additive and multiplicative phenomenological models is their emergence from simple and natural mechanistic models of transcriptional regulation. For instance, additive behavior naturally emerges from a model in which transcription factors can independently recruit polymerase to the promoter (*Scholes et al., 2017*; *Bothma et al., 2015*; *Bender et al., 2012*). Specifically, if signal A and signal B each induce the binding of different transcription factors to the enhancers of gene X, and these each independently result in an increased rate of binding of the polymerase to the promoter then the total rate of binding would be the sum of the two independent contributions. (This additive prediction assumes that the binding events are not so frequent as to saturate the promoter.) Consistent with this behavior, the deletion of pairs of enhancers at the mouse β-globin locus resulted in additive reductions in gene expression (*Bender et al., 2012*), and CRISPRa-based activation of enhancer subsets resulted in additive increases in gene expression for

several genes in an endometrial cancer cell line (*Ginley-Hidinger et al., 2019*). However, these experiments are typically limited to small sets of genes, making it difficult to conclude that additive behavior is the default, and indeed deviations from additive behavior are prevalent (*Bothma et al., 2015*; *Ginley-Hidinger et al., 2019*; *Scholes et al., 2019*).

Another oft-cited phenomenological observation is multiplicative integration of two transcriptional signals. One common model that can readily explain multiplicative integration is the so-called 'thermodynamic model', in which it is assumed that equilibrium binding levels of RNA polymerase to the promoter is the control point for transcriptional regulation (*Ackers et al., 1982*; *Bintu et al., 2005*; *Phillips et al., 2019*; *Scholes et al., 2017*; *Sherman and Cohen, 2012*). In a simple instantiation with two transcription factors, A' and B', which mediate the effects of signals A and B on gene X, each factor individually lowers the binding energy of RNA polymerase to the promoter, increasing its affinity (*Bintu et al., 2005*). If both transcription factors are present, then the changes in binding energy add, and hence, given that the probability of a transcription factor recruiting RNA polymerase II depends exponentially on binding energy, the net change in equilibrium binding levels of RNA polymerase II would multiply. Multiplicative activation by two RNA polymerase-binding factors has been seen in mutant *E. coli* experiments after λcl- and CRP-binding sides were placed adjacent to a *lacZ* promoter (*Joung et al., 1994*). In eukaryotes, thermodynamic models have been successful in predicting how engineered combinations of a few known transcription-factor-binding sequences next to a promoter affect the transcription of reporter genes in yeast and mouse embryonic stem cells, explaining ~50% of the variance in reporter gene expression, and up to 72% of the variance when non-multiplicative interaction terms are included (*Fiore and Cohen, 2016*; *Gertz et al., 2009*). However, it is unclear from many of these assays, most of which focus on promoter manipulations, how prevalent and general the multiplicative predictions of the simplest version of the thermodynamic model are, especially given that many combined responses are known to follow more simple additive predictions.

While potential mechanisms underlying additive and multiplicative behavior are straightforward, there is no a priori reason to believe that most genes would follow one or the other, or either at all. Indeed, a larger class of 'kinetic' models of transcription (which represent transcription as a coupled series of chemical reactions with distinct signal-responsive rates) have been shown to admit a wide variety of behaviors, ranging from sub-addition to super-multiplication (*Scholes et al., 2017*). A systematic test of these different phenomenological types of combined responses has yet to be done, in part because there is a lack of transcriptome-wide experiments in the literature that treat cells with two signals both individually and in combination. (A notable exception is *Goldstein et al., 2017*, where the authors use dual-signal treatment and a heuristic approach to find synergistic and antagonistic genes but do not compare underlying phenomenological models of combined responses.) Thus, it remains unknown if combinatorial gene regulation is primarily additive, multiplicative, or a wide distribution of everything in between (and beyond).

Upstream of transcription, it is also unclear how multiple signals coordinately affect transcription factor binding activity at *cis*-regulatory elements. For instance, if each signal results in the binding of a specific set of transcription factors at a particular regulatory region individually, then do these two different sets of factors bind with the same probability when both signals are applied? Or are these probabilities affected by potential regulatory interactions between the signals? And how might these binding probabilities and potential interactions affect expression of the target genes? There is only limited transcription-factor-binding data available for experiments where cells receive multiple signals simultaneously (*Goldstein et al., 2017*), and then using ChIP-seq, which only reports binding profiles for specific transcription factors. Pairing combined response experiments with chromatin accessibility measurements, which correlate with aggregate transcription factor binding data (*Thurman et al., 2012*), has the potential to answer these questions in a more comprehensive manner than ChIP-seq would allow for.

Experimentally, part of what makes it difficult to compare phenomenological models of combined responses is that additive and multiplicative models can give nearly indistinguishable predictions, especially when one or both of the signals' effects are relatively small. As such, often experimental data will be consistent with, say, a multiplicative or additive model (or weighted variants of such models), but it is difficult to exclude the possibility of the other model, especially when only a limited number of genes are considered (*Rothschild et al., 2014*; *Kaplan et al., 2008*; *Geva-Zatorsky et al., 2010*; *Rapakoulia et al., 2017*). With current genome-wide expression profiling

tools, however, it may be possible to query the integration modes of sufficiently many genes so as to discriminate between additive, multiplicative, and other phenomenological model predictions for at least some subset of genes, thus enabling a larger scale view of gene regulation's tendencies toward specific combined response behaviors.

Here, we profiled MCF-7 cells with paired RNA-seq and ATAC-seq measurements after we exposed them to retinoic acid, TGF-β, and both signals. We found that while genes' transcriptional responses exhibit a wide variety of behaviors when combining these two signals, they generally tended toward either addition or multiplication. ATAC-seq peaks, on the other hand, appeared to prefer addition as the default operation for combining two signal effects, although a minority of peaks clearly showed sub-additive or super-additive behavior. Genes with super-additive ATAC-seq peaks nearby were more likely to have a multiplicative or super-multiplicative transcriptional responses to retinoic acid and TGF-β. These data provide a comprehensive and systematic view of transcriptional responses to combined signal treatments.

## Results

### Upregulated genes gravitate toward addition and multiplication when combining the transcriptional effects of both signals

To quantitatively measure how gene regulation depends on multiple input signals, we performed three replicates of a paired RNA-seq and ATAC-seq experiment using MCF-7 cells (human breast carcinoma; selected for being well-characterized in its response to the two signals chosen). Prior to sequencing, we treated these cells with three different doses of TGF-β (1.25, 5, and 10 ng/ml), retinoic acid (50, 200, and 400 nM), or both signals (low, medium, and high dosages of both TGF-β and retinoic acid simultaneously) for 72 hr (*Figure 1B*). We waited 72 hr to create a larger set of differentially expressed genes to use in subsequent analyses, and chose doses that led to broad changes in transcription and chromatin accessibility (*Figure 1B*; see Materials and methods for discussion of doses chosen). Initial analysis showed that the number of differentially expressed genes and differential peaks increased in a dose-dependent manner, and that all genes that were upregulated in both individual signal treatments were also upregulated in the combination treatment (*Figure 1B*). We focused our analysis on upregulated genes and upregulated ATAC-seq peaks due to their greater dynamic range in effect sizes and their more straightforward interpretation in the context of potential binding of activators to increase the transcription of nearby genes. (Note that our ethanol 'vehicle' controls were performed at three different cell concentrations, and there were no significantly differentially expressed genes between concentrations. We did not, however, add the signals to different concentrations of cells or cells at different points in the cell cycle, in which context the signals may exert differential effects.)

We defined a master set of 1398 genes by selecting the set of genes that were significantly upregulated in any dose of the combination treatment (log2 fold-change $\geq 0.5$ and Benjamini-Hochberg adjusted p value $\leq 0.05$) and that had increased expression in all doses of each individual signal (*Figure 1D*). If we had selected the full set of all genes upregulated in any dose of the combined treatment, we would have analyzed a set of 2246 genes (*Figure 1D*). We required the change in expression to be positive for both individual signals, however (i.e. $\Delta_A > 0$ and $\Delta_B > 0$), in order to maintain a consistent mapping between our categorical description of combined responses (e.g. 'sub-additive', 'super-multiplicative' (*Figure 1—figure supplement 1*)) and our continuous 'c value' description of combined responses defined in Appendix 1 and *Figure 1A*. Requiring $\Delta_A > 0$ and $\Delta_B > 0$ in our master set of genes was necessary to guarantee that sub-additive combined transcriptional responses always had c values less than 0 and that super-multiplicative responses always had c values greater than 1. Imposing the conditions of $\Delta_A > 0$ and $\Delta_B > 0$ removed 37.8% of the 2246 genes that showed a significant increase in expression in the combined treatment (*Figure 1D*), leaving 1396 of the 1398 genes that ultimately fed into our analyses. Inclusion of genes with negative changes after individual signal treatments would require a more elaborate analysis framework to encompass the much larger variety of categorizations of potential responses that would be difficult to characterize with the number of genes in our analysis. (There were only two genes that were significantly downregulated in the combined treatment while also having $\Delta_A > 0$ and $\Delta_B > 0$ at all doses

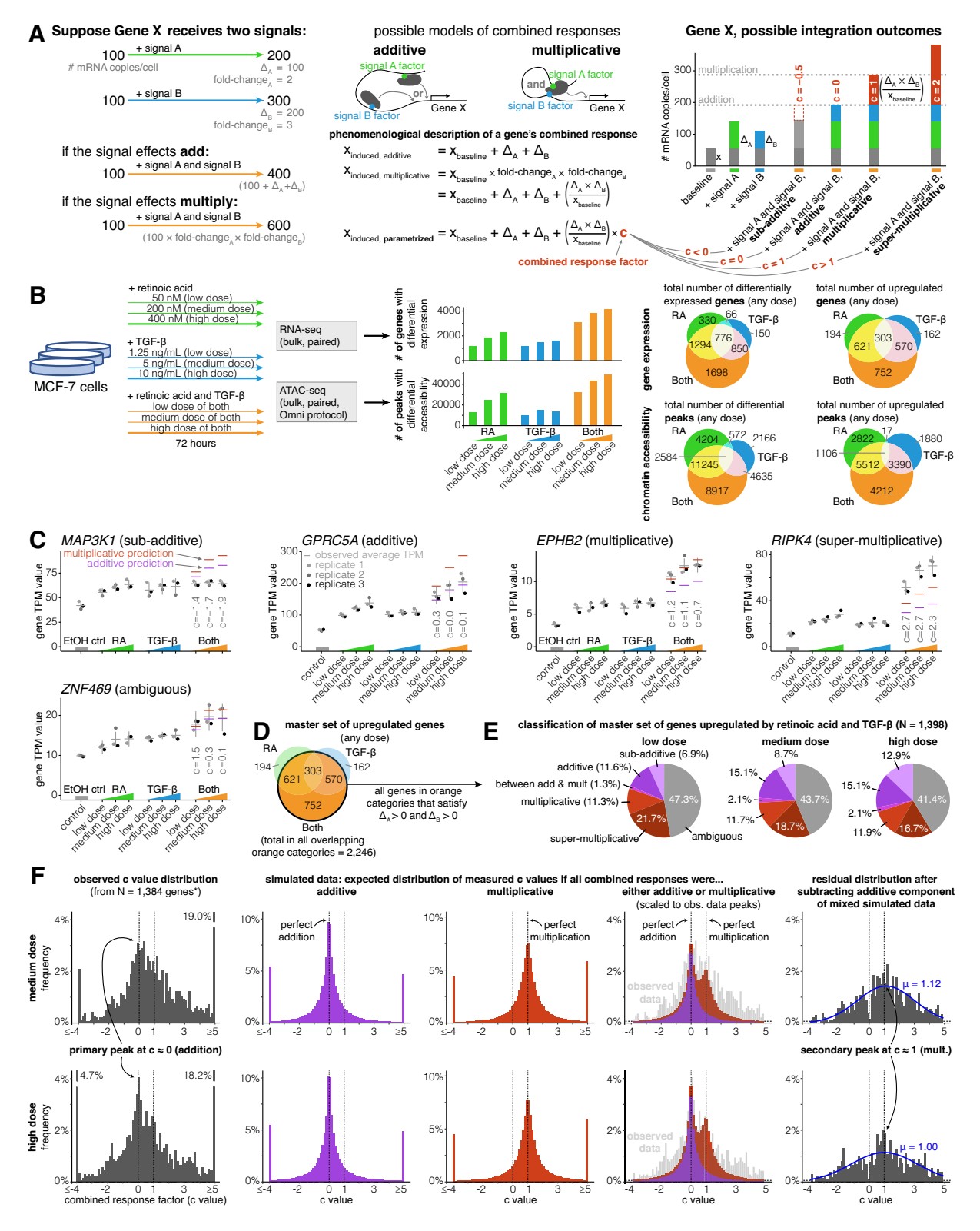

**Figure 1.** Addition and multiplication are enriched modes of signal integration in upregulated genes. (**A**) Example of additive vs. multiplicative effects on expression of hypothetical gene X, mathematical formulation of the combined response factor, and illustration of how the value of the combined response factor (c value) reflects whether a combined gene expression response is sub-additive, additive, multiplicative, or super-multiplicative. (**B**) Schematic of signal response experiments in MCF-7 cells. Briefly, we treated MCF-7 cells with three different dosages of retinoic acid, TGF-β, or both

*Figure 1 continued on next page*

*Figure 1 continued*

signals for 72 hr, then performed bulk RNA-seq and ATAC-seq at the endpoint. We show the number of differentially expressed genes and peaks for each dose of each condition as well as the overlap between the sets of differentially expressed genes and differential peaks. (C) Five example genes representing sub-additive to super-multiplicative combined transcriptional responses, where we show each gene's transcripts per million (TPM) value for each replicate after single or combined signal treatments. Horizontal gray bars show the average TPM value, and error bars represent the 80% confidence interval of the estimated underlying Gaussian distribution of each dosage and condition (see Materials and methods for parameter estimation details). (D) Illustrated definition of master set of upregulated genes. (E) Frequency of each type of combined response behavior for each dosage in the master set of genes. (F) Simulated, observed, and residual histograms of c value distributions for the medium and high doses. In the simulated mixture model, we randomly simulated combined responses to be either additive or multiplicative based on the relative frequency of additive vs. multiplicative combined transcriptional responses that we observed at each dose in 1E. Annotated percentages at broken bars represent the fraction of c values in the tail beyond the limits of the x axis of the graph. *For all c value analyses, 14 genes with a control TPM of zero were removed from the master set of genes, as they end up misleadingly having c values of exactly 0 regardless of the effects of retinoic acid and TGF-β. The online version of this article includes the following figure supplement(s) for figure 1:

**Figure supplement 1.** Explanatory schematics for model of gene expression variation, classification of combined responses, and simulating new additive or multiplicative combined responses.

**Figure supplement 2.** TGF-β, and not retinoic acid, leads to an increase in nuclear pSMAD2 levels in MCF-7 cells.

**Figure supplement 3.** Nuclear retinoic acid receptor alpha levels are stable across treatment conditions.

**Figure supplement 4.** A secondary peak occurs at or near perfectly multiplicative combined transcriptional responses (c = 1) after subtracting a distribution of simulated additive responses from the observed distribution of c values.

**Figure supplement 5.** The combined response factor tends to remain stable or decrease with increasing signal dosage.

of each individual signal treatment; we elected to also include these two genes in our master set for the total of 1398.)

In our analysis of combined transcriptional responses, we assumed that retinoic acid and TGF-β exhibited their effects on common target genes through distinct transcription factors. To justify this assumption, we confirmed that there was little cross-activation of pSMAD2 (which serves as a proxy for the readout of TGF-β signaling) by performing immunofluorescence targeting pSMAD2 upon the addition of TGF-β and retinoic acid individually (*Figure 1—figure supplement 2A–B*). We saw that TGF-β treatment rapidly increased the nuclear signal of pSMAD2 (by 40 min), which remained above baseline until the final time point at 72 hr, whereas retinoic acid treatment induced no changes in pSMAD2 signal relative to baseline (*Figure 1—figure supplement 2C–E*). Nuclear expression of retinoic acid receptor alpha, which resides in the nucleus regardless of activation level (*Mangelsdorf and Evans, 1995*), was stable between conditions at all time points (*Figure 1—figure supplement 3*). Subsequent transcription factor motif analysis of our ATAC-seq data, however, suggested that retinoic acid receptor alpha (RARA) is activated by retinoic acid and not TGF-β (see section titled 'Motif analysis reveals that sub-additive peaks have a depletion of AP-1 and an enrichment of CTCF motifs while super-additive peaks have an enrichment of SMAD motifs'). This same motif analysis also suggested that retinoic acid and TGF-β largely increased the activity of distinct transcription factors at the 72 hr time point, meaning that the secondary effects of retinoic acid and TGF-β are likely mediated through the activity of distinct transcription factors.

Within our master set of 1398 upregulated genes, we found a variety of different combined transcriptional response behaviors ranging from sub-addition to super-multiplication (*Figure 1D–F*). A transcriptional response is additive when the combined treatment effect represents the sum of the individual treatment effects, and multiplicative when the combined treatment represents the product of the individual treatment fold-changes. When both signals upregulate the expression of a gene, a multiplicative response is always higher than an additive response (Appendix 1; *Figure 1A*). To systematically classify the combined transcriptional responses at each gene, we used a statistical approach where we assumed each observation of a gene's expression value was derived from a Gaussian distribution (see Materials and methods). We classified a combined transcriptional response as sub-additive, additive, multiplicative, or super-multiplicative by comparing where a 'perfect' hypothetical additive or multiplicative response lay with respect to the 80% confidence interval of the combined treatment's expression value (*Figure 1—figure supplement 1B*). If both the hypothetical additive and the hypothetical multiplicative predictions lay within the confidence interval, we classified the response as ambiguous (*Figure 1—figure supplement 1B*). Using this approach, we found that at the medium dose, 8.7% of genes had sub-additive combined transcriptional responses,

15.1% had additive responses, 2.1% had between an additive and multiplicative response, 11.7% had multiplicative responses, 18.7% had super-multiplicative responses, and 43.7% had ambiguous responses (*Figure 1D*), suggesting that there is no single dominant category of combined response behavior. However, while the categories of addition and multiplication are appealing due to their correspondence to these simple phenomenological models, there is no a priori reason to believe that all or even most genes should necessarily adhere to either of these possibilities.

In order to quantitatively describe the combined transcriptional response characteristics of any gene without any presupposition of additive or multiplicative behavior, we defined a continuous parameter, hereby referred to as a gene's combined response factor or 'c' value, that places the gene in an exact location on the spectrum of possible combined response behaviors (Appendix 1; *Figure 1A*). We could then solve for any gene's c value (within experimental error) after measuring the individual signal effects and the combined treatment effect. For an upregulated gene, a c value of 0 would indicate perfect addition, a c value of 1 indicates perfect multiplication, a c value less than 0 indicates sub-addition, and a c value greater than one indicates super-multiplication (see *Figure 1A* for equation). We wondered what the distribution of c values would look like across our master set of upregulated genes, and whether this distribution would tell us anything about genes' natural inclinations for specific combined response behaviors. For instance, if this distribution had its main peak at c = 0.5, it would imply that genes naturally prefer to integrate two signals in a manner that lies between addition and multiplication. At all doses of combination treatment, we observed a wide peak centered around c = 0 (additive), with a hint of a secondary peak at c = 1 (multiplicative), suggesting that the integration of the effects of two signals is preferentially additive or multiplicative (*Figure 1F*; *Figure 1—figure supplement 4*).

In order to more rigorously demonstrate the preferences for these two values of c, we performed a series of simulations and statistical analyses. First, we generated simulated data taking into account measurement noise to estimate what the expected distributions of c would look like if signal integration was wholly additive or multiplicative. For each gene, we made three random draws for expression levels in both signal conditions based on the actual expression measurements and variance of those measurements to mimic our actual data (*Figure 1—figure supplement 1C*). We then computed what we would have measured c to be based on these simulated measurements. This 'null' produced broad peaks centered around c = 0 and c = 1, respectively, and a superposition of these two nulls appeared to match our experimentally measured distribution of c values (*Figure 1F*). In order to more clearly demonstrate the existence of a secondary peak at c = 1, we subtracted off from the distribution a purely additive null model (as computed above, fit to the observed distribution). The resultant residual distribution was a broad peak centered roughly around c = 1 (a Gaussian fit to the residual gave a fit centered at c = 1.12 and c = 1.00 at medium and high doses, respectively), consistent with our multiplicative simulated data (*Figure 1F*; *Figure 1—figure supplement 4A*). We showed that this residual distribution was not likely to be due to statistical fluctuations by computing a p-value for the possibility of obtaining as big a residual in a sliding window by random chance (*Figure 1—figure supplement 4B*). Overall, while there is the possibility of further peaks within our data, our data most strongly support the existence of two peaks in the c-value histogram, one corresponding most closely with an additive model, and the other with a multiplicative model.

While our superimposed distribution of c values derived from simulated additive and multiplicative combined responses bears a close resemblance to our observed distribution of c values in the neighborhoods of c = 0 (addition) and c = 1 (multiplication), the tails of the observed c value distribution are clearly heavier (*Figure 1F*). These heavier tails illustrate that biological variation, rather than measurement error, produces a significant amount of sub-additive (c < 0) and super-multiplicative (c > 1) combined transcriptional responses.

We next wondered how a gene's combined response factor (c value) depended on dosage of the input signals. In theory, the c value might remain stable as dosage increases, monotonically increase or decrease as dosage increases, or may appear to be 'random' with respect to dose, perhaps due to complex unobserved dose-dependent gene regulatory interactions. To distinguish between these possibilities, we plotted how a set of upregulated genes' c values changed as they moved from low to medium to high dose of combination treatment with retinoic acid and TGF-β (*Figure 1—figure supplement 5B–D*). To generate a subset of reliable c value estimates within our master set of genes, we selected genes for which $\frac{\Delta_A \Delta_B}{x_{baseline}} \geq 2$ transcripts per million (TPM) and $\frac{\Delta_A \Delta_B}{x_{baseline}} \geq x_{baseline}$

(*Figure 1—figure supplement 5C*). Since $\frac{\Delta_A \Delta_B}{x_{baseline}}$ captures the difference between the multiplicative and additive predictions, the estimation of c is more reliable when $\frac{\Delta_A \Delta_B}{x_{baseline}}$ is large, because when that number is large it is less susceptible to technical variability. We found that most genes' c values were stable or moderately decreased with increasing signal dose, suggesting that the function a gene uses to combine two signals is mostly stable, with a tendency towards 'saturation' with increasing dose (i.e. the function itself moves in the direction of sub-additivity when dosage increases).

## Increases in chromatin accessibility are largely additive

Transcriptional regulation is thought to occur largely via the binding of transcription factors, but it remains unknown how the transcription factors associated with the effects of individual signals might interact upon the addition of both signals simultaneously. We performed ATAC-seq on the same populations described earlier, reasoning that the observation that changes in chromatin accessibility have been shown to correlate with changes in aggregate transcription-factor-binding activity (*Thurman et al., 2012*) meant that we could infer something about transcription factor binding at these sites. Note that the extent to which changes in chromatin accessibility quantitatively reflect changes in transcription factor occupancy is currently unknown, and may depend on the mechanism by which binding of transcription factors leads to opening of chromatin, such as displacement of nucleosomes by pioneer factors, recruitment of secondary transcription factors, or recruitment of chromatin remodeling complexes (*Zaret and Carroll, 2011*; *Klemm et al., 2019*). Reassuringly, our initial motif enrichment analysis revealed that retinoic acid receptor alpha (RARA) and three TGF-β pathway transcription factor motifs (SMAD3, SMAD4, and SMAD9) were highly enriched in their respective individual signal treatment conditions (*Figure 4—figure supplement 1B*). Note that our motif analysis also indicated some degree of activation of RARA by TGF-β and some degree of activation of SMAD3 and SMAD9 by retinoic acid, which led to even higher enrichment levels of these factors in the combined treatment condition (*Figure 4—figure supplement 1A*). We did not, however, observe cross-activation of pSMAD2 by retinoic acid in immunofluorescence experiments (*Figure 1—figure supplement 2*).

We then wondered how well simple additive and multiplicative phenomenological predictions corresponded to the increase in chromatin accessibility at upregulated peaks in the combined treatment. We found that an additive model was generally highly predictive and matched the observed increases in ATAC-seq fragment counts more accurately than the multiplicative model; the multiplicative model generally predicted larger changes in accessibility than we experimentally observed (*Figure 2—figure supplement 1*). To quantify the degree to which the additive prediction was accurate, we defined a new metric, the fold-change difference in accessibility from an additive model prediction, hereby referred to as a peak's 'd' value, to create a distribution that illustrates the extent to which the size of a peak in the combination treatment condition deviated from additive model predictions (*Figure 2A–B*). We found that at upregulated peaks, our observed distribution of d values was centered at zero, highlighting how addition appears to be the 'default' operation at upregulated peaks (*Figure 2C*). This default additive behavior may correspond to a mechanistic model in which each signal stimulates an independent set of chromatin-opening transcription factors that independently and rarely bind DNA (*Figure 2E*).

Given the general accuracy of the additive model for upregulated peaks, we wondered to what extent deviations from additive model predictions represented true deviations as opposed to just measurement error. We produced randomly generated simulated data that matched the statistical properties of our actual data, assuming that the combined treatment would result on average in perfectly additive peak sizes (see Materials and methods for details). We found that our observed data are more widely dispersed than the simulations, indicating that a fair number of peaks are significantly sub-additive or super-additive (*Figure 2C*). We found that 19% of peaks were sub-additive and 16% of peaks were super-additive when we considered additive peaks to be those where a perfectly additive prediction lied within the 80% confidence interval of the measured peak fragment counts (*Figure 2D*). Thus, most upregulated ATAC-seq peaks displayed additive or near-additive combined responses, but significant fractions of peaks also displayed both sub-additive and super-additive combined responses.

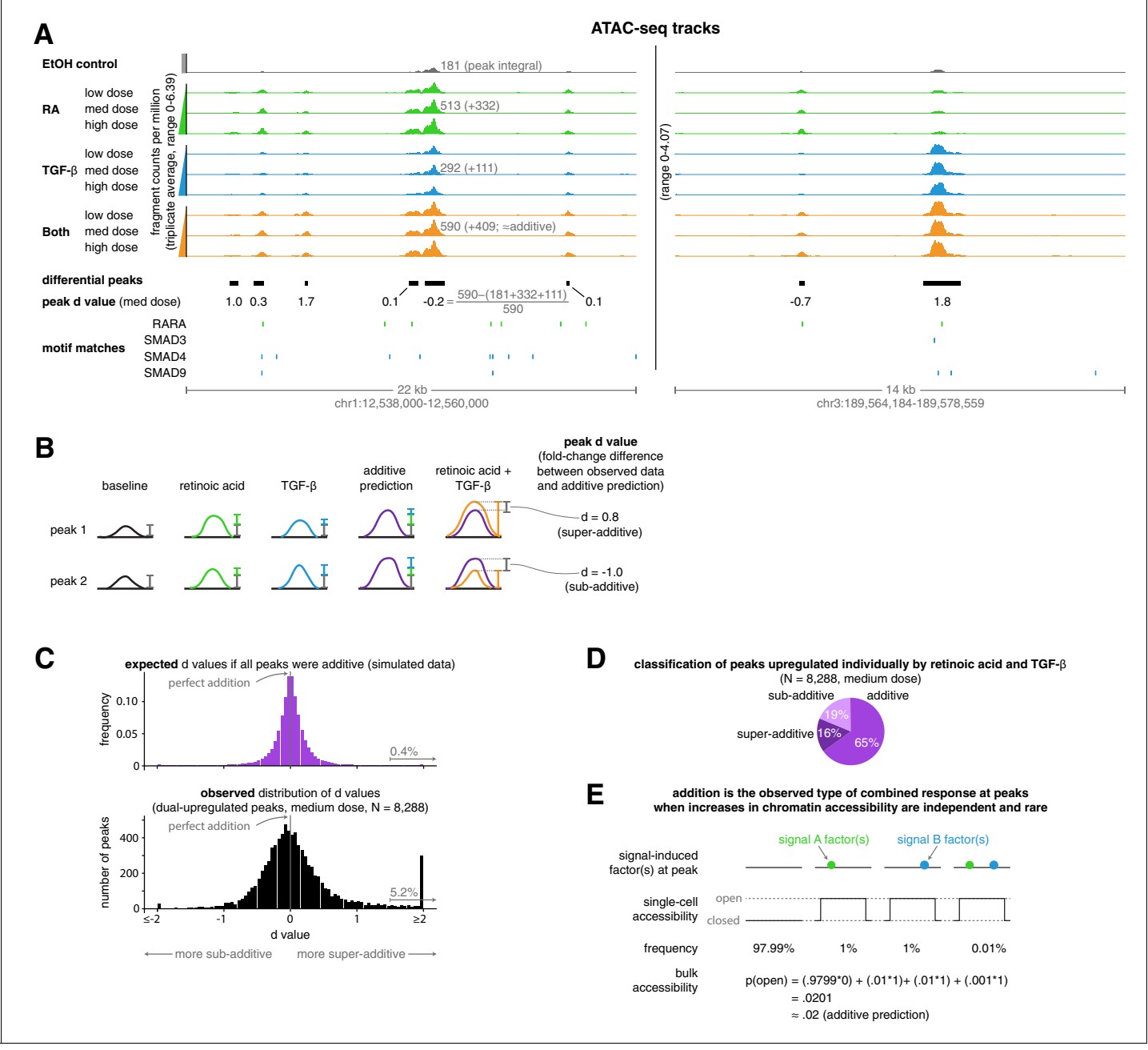

**Figure 2.** Addition is the default operation at upregulated differential peaks. (A) Example tracks of ATAC-seq data. Tracks illustrate the ATAC-seq fragment counts per million, with each value representing the average number of fragment ends per million within 75 bp of a given genomic coordinate. Annotated peak values represent the peak integral (the total number of normalized fragment counts measured within the peak), which we use to calculate the peak's d value. (B) Schematic illustrating examples of two peak's d values, where each d value represents the fold-change difference between the measured number of ATAC-seq counts in the combination treatment and the predicted number of ATAC-seq fragment counts when using an additive model. (C) Expected vs. observed distributions of the fold-change difference from an additive prediction for each peak. We generated the expected distribution by simulating 10 new observations for each peak from the distributions we estimated our original upregulated peaks to have come from, setting the mean of the combined treatment to a perfectly additive prediction (Materials and methods). (D) Classification of ATAC-seq peaks that were upregulated individually by retinoic acid and TGF-β. We considered a given peak to be additive when the additive model prediction lied within the 80% confidence interval of our estimated distribution of the given peak's normalized fragment counts in the combined treatment condition. (E) Schematic illustrating how combined binding responses may be additive when transcription factor binding is independent and rare.

The online version of this article includes the following figure supplement(s) for figure 2:

*Figure 2 continued on next page*

*Figure 2 continued*

**Figure supplement 1.** The combined response of peaks upregulated individually by retinoic acid and TGF-β is more consistent with an additive model than a multiplicative model.

## Super-additive peaks and pairs of individual signal-dominant peaks are more likely to be found near genes with multiplicative transcriptional responses

We next wondered if we could uncover the patterns of *cis*-regulatory element activity that may dictate how a gene's regulatory behavior would encode the observed integration of the transcriptional effects of two signals. We reasoned that the number of upregulated ATAC-seq peaks near a gene or the manner in which the nearby peaks themselves integrated the two signals' effects may predict the gene's combined transcriptional response behavior. For each transcriptionally upregulated gene, we counted the number of sub-additive, additive, and super-additive ATAC-seq peaks within 100 kb of its transcription start site. We found that, on average, genes that were transcriptionally additive had 2.7x more sub-additive ATAC-seq peaks nearby than genes with multiplicative transcriptional responses (medium dose, p=0.0012). Genes with multiplicative and super-multiplicative transcriptional responses had 2.5x or 2.6x, respectively, more super-additive ATAC-seq peaks nearby than genes with additive transcriptional responses (*Figure 3A*, medium dose, p=0.0016 or p=0.00016, respectively). Genes with multiplicative transcriptional responses also had more additive ATAC-seq peaks nearby than every other combined transcriptional response behavior at each dose we tested, with 1.3x more additive peaks than genes with additive transcriptional responses (*Figure 3A*, medium dose, p=0.12 compared to additive transcriptional responses, p=0.00089 compared to ambiguous transcriptional responses). The most prominent effect in this analysis was the observation that super-additive peaks are more likely to be near genes with multiplicative and super-multiplicative transcriptional responses, suggesting that cooperative interactions between transcription factors at neighboring enhancers may increase the expression of a gene when both signals are added together, that is, the gene's combined response factor.

When both signals affect accessibility at the same region of DNA, interactions between each signal's induced transcription factors and associated complexes can make it difficult to discriminate between mechanistic models of how transcription factors interact to regulate transcription. However, if the transcription factors affected by retinoic acid or TGF-β bind to distinct regions of DNA around the same gene, then there are likely no interactions between induced transcription factors and one can in principle discriminate between a simple thermodynamic model (prediction: multiplicative transcriptional effects) and an independent recruitment model (prediction: additive transcriptional effects). To increase the likelihood of selecting retinoic acid and TGF-β-exclusive transcription-factor-binding events, we searched near genes for upregulated peaks that responded exclusively to either retinoic acid or TGF-β. (We defined 'exclusive' here to mean that the peak size increase for a single signal was ≥90% that of the sum of the absolute peak size changes from both individual signals. Note that to generate a sufficiently large sample, we had to allow the selected genes to have non-exclusive peaks nearby as well because only 8.0% of gene-adjacent differential peaks met this exclusivity criteria for retinoic acid and only 3.4% met this criteria for TGF-β.) We then considered how likely genes with different combined transcriptional response behaviors were to have at least one retinoic acid-dominant and one TGF-β-dominant peak nearby (<100 kb to the transcription start site). We found that at each dose, genes with multiplicative transcriptional responses were the most likely to have at least one retinoic-acid-dominant and one TGF-β-dominant upregulated peak nearby (*Figure 3B*; 2.4x increase compared to genes with additive transcriptional responses at high dose, p=0.044), suggesting that the effects of independently upregulated peaks are most likely to act together to multiplicatively regulate transcription, which is more consistent with the predictions of the thermodynamic model.

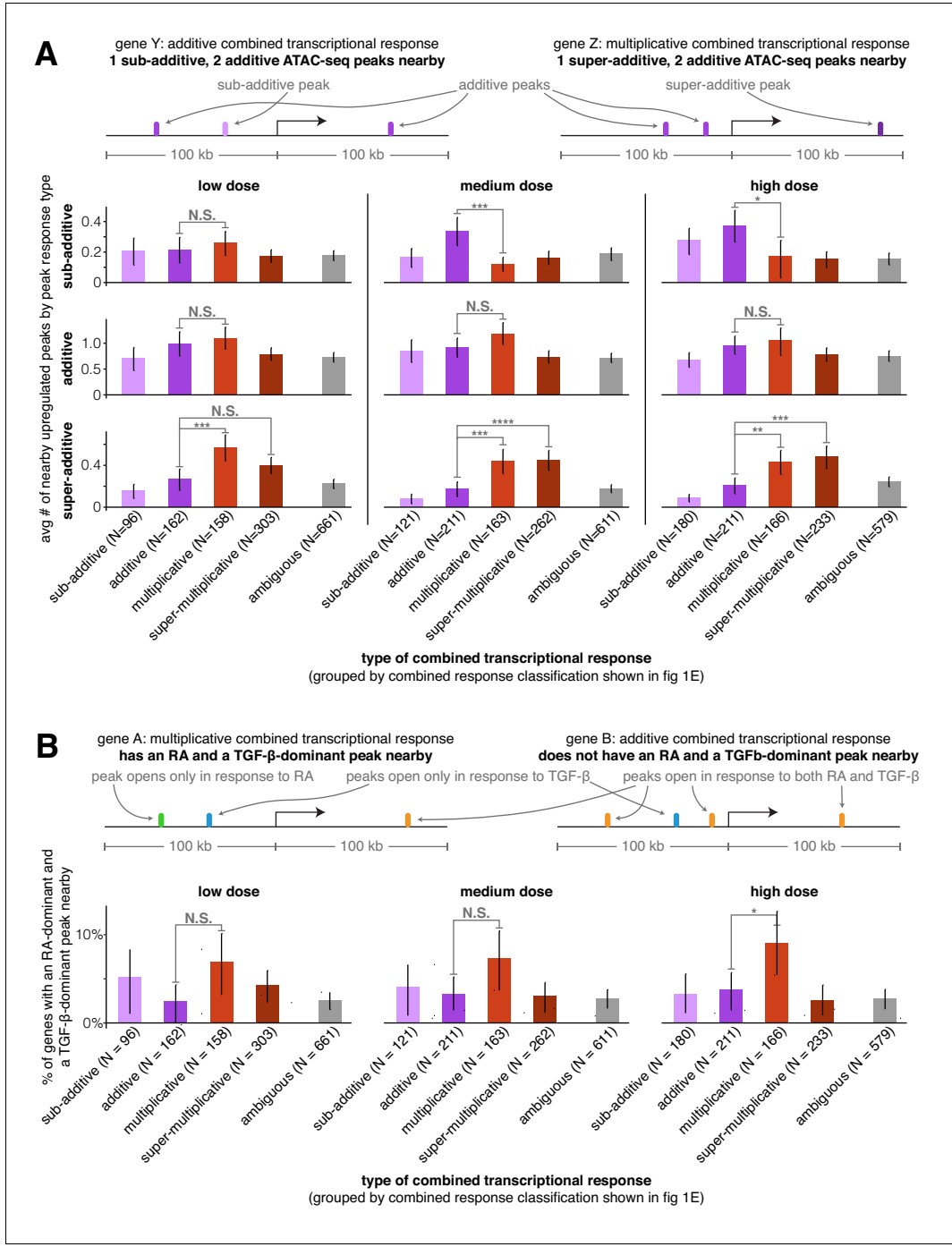

**Figure 3.** Super-additive ATAC-seq peaks are enriched near genes with multiplicative and super-multiplicative combined transcriptional responses. (**A**) For each type of combined gene expression response, we show the average number of upregulated sub-additive, additive, and super-additive ATAC-seq peaks within 100 kb of the gene's transcription start site. (**B**) For each combined transcriptional response behavior, we show the percentage of genes that have at least one peak that responds exclusively to retinoic acid and at least one peak that responds exclusively to TGF-β (where both peaks must lie within 100 kb of the gene's transcription start site). For an upregulated peak to be considered a mutually exclusive response, the change in ATAC-seq fragment counts in the individual treatment condition must be at least 9x larger in the major signal effect than the minor signal effect. *p<0.05; **p<0.01; ***p<0.002; ****p<0.0002. All p values were calculated using Student's t-test. All error bars represent the 90% confidence interval estimated using 10,000 empirical bootstrap samples of the peak sets used in each analysis.

## Motif analysis reveals that sub-additive peaks have a depletion of AP-1 and an enrichment of CTCF motifs while super-additive peaks have an enrichment of SMAD motifs

We next wondered if the activity of particular transcription factors was associated with combined increases in chromatin accessibility that were either sub-additive, additive, or super-additive. To approach this question, we first identified a set of the 50 transcription factors with the largest predicted changes in activity in our full set of differential peaks using the chromVAR package and its associated curated cisBP database of transcription factor motifs (*Schep et al., 2017*). These factors included the canonical retinoic acid and TGF-β effectors RARA, SMAD3, SMAD4, and SMAD9, as well as forkhead box factors and ETS family factors (enriched in the retinoic acid condition), AP-1 factors (enriched in the TGF-β condition), and HOX and NF-κβ factors (enriched in both the retinoic acid and TGF-β conditions). We manually added the CTCF motif to this set of enriched motifs to see if putative insulators behaved differently than other *cis*-regulatory elements. For each of these transcription factors, we calculated a motif enrichment score in each condition (based on the bias-uncorrected deviation score from chromVAR) that represents the percentage change in ATAC-seq fragment counts in all peaks that contain the given transcription factor's motif (*Figure 4A*). For example, the motif enrichment score of 0.19 for RARA in the retinoic acid condition means that peaks containing RARA motifs saw an average increase of 19% in ATAC-seq fragment counts after retinoic acid treatment (note that to decrease the variability of motif enrichment score estimates, we pooled together the low, medium, and high doses for each condition). Retinoic acid and TGF-β treatment thus led to activation of both distinct and shared transcription factor families, with combination treatment showing similar activation of distinct factors and higher activation of shared factors (*Figure 4A*).

We then tested if any of the transcription factor motifs we identified were more enriched in sub-additive or super-additive peaks compared to additive peaks. Because sub-additive peaks were on average 8% narrower and super-additive peaks were on average 36% wider than additive peaks (*Figure 4B*), we compared the number of motif matches found per 150 bp of each peak type. When compared to additive peaks, sub-additive peaks showed 21% fewer total motif matches per 150 bp in our set of enriched motifs (p=6.6e$^{-14}$) and 7% fewer total motif matches per 150 bp when using the entire cisBP database (p=1.5e$^{-8}$), suggesting that sub-additive peaks are slightly depleted for motifs overall while being even more depleted for the motifs in our enriched set (*Figure 4C*). Sub-additive peaks were especially depleted for SMARCC1 motifs (0.6x the motif density of additive peaks, p=1.2e$^{-15}$) as well as AP-1 subunit motifs such as JUN (0.6x density, p=3.4e$^{-13}$) and FOS (0.6x density, p=6.2e$^{-13}$; *Figure 4E*). Sub-additive peaks did, however, show a strong enrichment of CTCF motifs, with 1.6x and 3.2x more motif matches per 150 bp than in additive and super-additive peaks, respectively (p=2.9e$^{-11}$ and p<2.2e$^{-16}$, respectively; *Figure 4E*), suggesting that insulator proteins like CTCF may attenuate the combined activity of signal-induced transcription factors or the chromatin remodeling complexes they may recruit.

Super-additive peaks generally had the same motif densities as additive peaks, with the exception of an increase in the density of SMAD motifs (1.8x, 1.4x, and 1.5x increase of SMAD3, SMAD4, and SMAD9 motif density compared to additive peaks; p=4.4e$^{-4}$, p=8.5e$^{-5}$, p=1.5e$^{-5}$) and a depletion of several ETS family factors (0.6x the motif density of additive peaks for ELF1, p=0.048; *Figure 4E*). The higher frequency of SMAD motifs in super-additive peaks suggests that SMAD transcription factors may interact with retinoic-acid-induced chromatin remodeling factors or retinoic-acid-induced transcription factors.

We next wondered how strong of an effect each motif had on its 'host' peak's tendency to have a super- or sub-additive combined response. To estimate this effect, we took each motif, found all peaks that contained that motif that were upregulated by both TGF-β and retinoic acid individually, and computed the deviation from the additive prediction (d value) (*Figure 4F*). Here, we found that the presence of SMAD or NF-κβ motifs resulted in the largest increases in a peak's tendency to be super-additive, possibly suggesting that SMAD proteins have one of the most potent interactions with a retinoic-acid-induced transcription factor or chromatin remodeling complex in our system. Note that since we observed that both retinoic acid and TGF-β led to increases in NF-κβ factor activity (*Figure 4A*), the increase in d value associated with NF-κβ motifs' could reflect synergistic

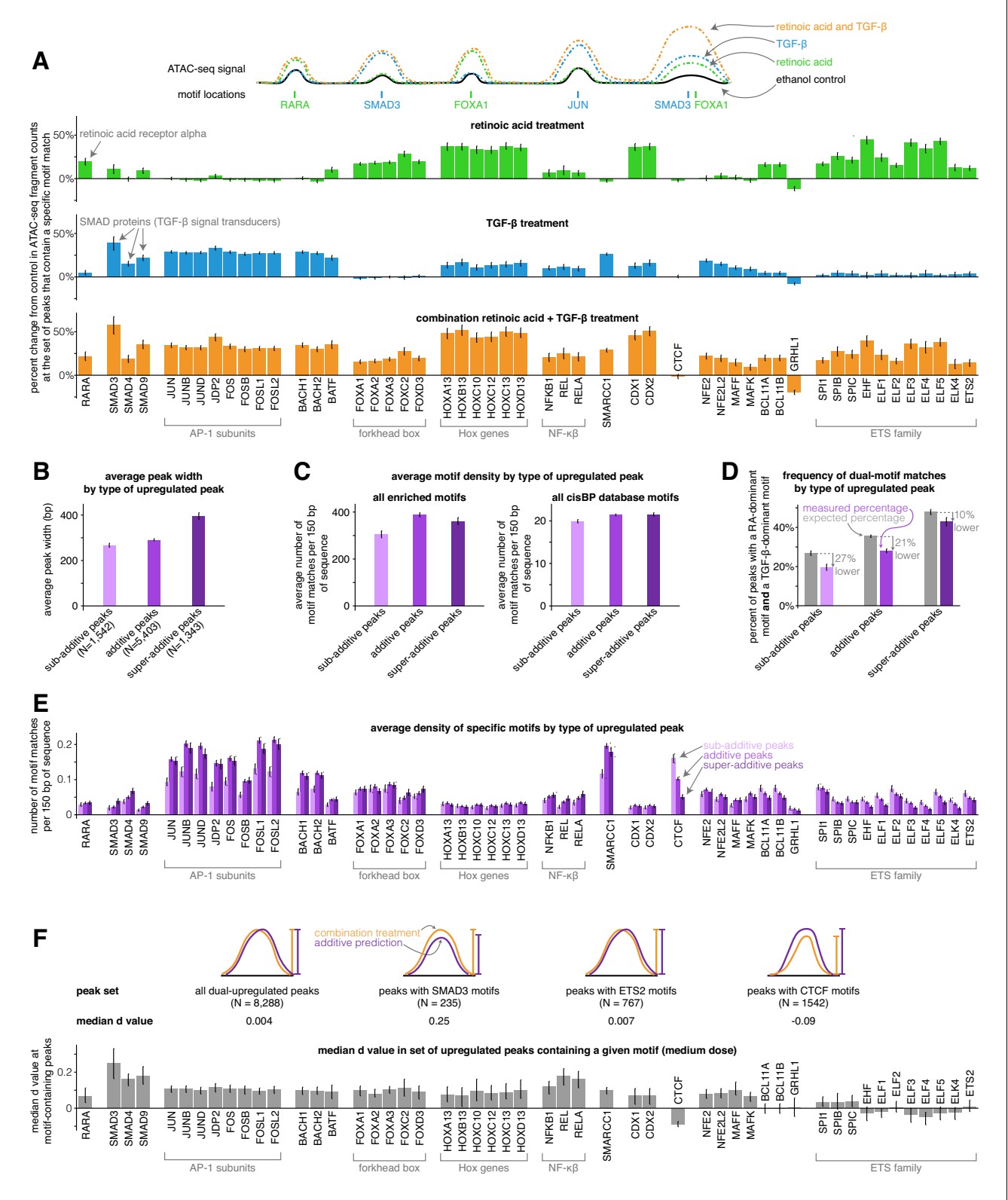

**Figure 4.** Sub-additive peaks are depleted for AP-1 motifs, enriched for CTCF motifs, while super-additive peaks are enriched for SMAD motifs. (**A**) Motif enrichment analysis in each condition for the top 50 most variable transcription factor motifs identified by chromVAR. (CTCF was manually added to this set, making the total 51). Y-axis represents the percentage change in ATAC-seq signal at motif-containing peaks compared to ethanol control samples. For each condition, we pooled together the replicates for each of the three dosages, resulting in nine replicates each for retinoic acid, TGF-β,

*Figure 4 continued on next page*

*Figure 4 continued*

and combination treatment. (B) Average peak width of peaks upregulated individually by retinoic acid and TGF-β by type of combined response. (C) Average motif density in each type of peak upregulated individually by retinoic acid and TGF-β, using the enriched motif set and the full cisBP database. (D) Expected vs. measured percentage of dual-motif matches (one retinoic-acid-dominant motif and one TGF-β dominant motif) for each type of upregulated peak. We calculated the expected percentage by randomly shuffling motif matches within each peak set (see methods for details). Error bars represent the 5th and 95th percentile of the null distribution for expected percentages and the 90% bootstrapped confidence interval for measured percentages. (E) Motif density by type of upregulated peak for each motif in our enriched set. (F) For a given enriched motif, the median d value at medium dose for all upregulated peaks that contain the motif (higher d values indicate more super-additivity in peaks containing a given motif; the median d value for all upregulated peaks was 0.004). All error bars (except for the error bars for expected percentages in D) represent the 90% confidence interval estimated using 1000 empirical bootstrap samples of the peak sets used in each analysis.

The online version of this article includes the following figure supplement(s) for figure 4:

**Figure supplement 1.** Canonical retinoic acid and TGF-β signaling motifs (RARA, SMAD3, SMAD4, SMAD9) are enriched in their respective signal treatment conditions.

activation of NF-κβ factors rather than cooperative interactions between NF-κβ factors and other induced transcription or chromatin remodeling factors.

We hypothesized that cooperative interactions between transcription factors may lead to super-additive increases in chromatin accessibility. To evaluate if our data supported this hypothesis, we tested if super-additive peaks were more likely to have both a retinoic-acid-enriched motif and a TGF-β-enriched motif. We defined retinoic acid-enriched factors to be retinoic acid receptor, FOX, and ETS-family factors, and we defined TGF-β-enriched motifs to be SMAD, AP-1, BACH, BATF, SMARCC1, NFE2, NFE2L2, MAFF, and MAFK factors. We found that all categories of peaks (including super-additive) were less likely to have dual-motifs than expected based on a null distribution we generated by randomly shuffling motif matches across peaks (*Figure 4D*, p<0.001 for sub-additive, additive, and super-additive peaks, see Materials and methods for null distribution details. The higher expected rates of dual motif matches may be explained by the fact that binding sites for the same transcription factor are often found in clusters *Gotea et al., 2010*; the motif shuffling process disperses these binding sites more evenly). Super-additive peaks were closer to their higher expected rate than sub-additive and additive peaks (with super-additive, additive, and sub-additive peaks having dual-motif match rates that were 10%, 21%, and 27% lower than expected, respectively). While the effect is modest, the relatively higher rate of dual-motif matches in super-additive peaks provides some support for the idea that peak super-additivity may result from cooperative interactions between retinoic acid and TGF-β transcriptional effectors.

## Discussion

Here, we have asked how cells respond transcriptionally to combinations of signals. In principle, the transcriptional response to such combinations could range over a spectrum of different possibilities, and the mechanistically motivated 'additive' and 'multiplicative' modes need not be favored. We were thus surprised to see that combined responses did seem to favor the simple additive and multiplicative phenomenological models.

Additive and multiplicative outcomes need not in principle be favored in any way. Mechanistic models of transcriptional regulation, in particular kinetic models, can yield a range of phenomenological predictions, spanning these two possibilities and more (*Scholes et al., 2017*). The primary reason behind the popularity of the independent recruitment model (which predicts additive behavior) and the thermodynamic model (which predicts multiplicative behavior) is their simplicity, hence our surprise. It is of course important to realize that just because the predictions of a particular mechanistic model match these experimental outcomes does not mean that there are not other models that may also match our experimental findings. Indeed, these simple models, which inherently posit that regulation acts via a single rate-limiting step, are incompatible with recent results demonstrating that regulation can act via multiple steps, and also typically have not been applied to complex regulatory mechanisms that involve long-range promoter-enhancer contacts (*Bartman et al., 2019*; *Blau et al., 1996*; *Fuda et al., 2009*; *Nechaev and Adelman, 2011*; *Stampfel et al., 2015*). Further combined theoretical and experimental work would be required to

determine the experimental signatures beyond simple additivity or multiplicativity that could distinguish such models from each other.

Although they were the minority of cases, we did observe a large number of sub-additive and super-multiplicative combined responses. Super-multiplicative combined responses may reflect cooperative interactions between retinoic acid and TGF-β-induced factors, in which binding of a retinoic acid factor to DNA strengthens the binding of a TGF-β factor to nearby DNA or vice versa. This type of interaction is consistent with our finding that super-multiplicative gene expression responses are associated with nearby super-additive ATAC-seq peaks (on the assumption that super-additivity of ATAC-seq peaks reflects cooperative binding of transcription factors to DNA) (*Figure 3A*). However, given that ATAC-seq peaks likely have additional routes to super-additive increases in accessibility (perhaps involving chromatin remodeling factors affected by our signals), further work would be needed to demonstrate that super-multiplicative transcriptional responses are indeed a result of direct binding interactions at enhancers. Sub-additive transcriptional responses have been proposed to reflect saturation of cis-regulatory elements (*Bothma et al., 2015*; *Scholes et al., 2019*). Saturated cis-regulatory elements would in principle show up as sub-additive ATAC-seq peaks in our analysis, but we did not observe an increase in sub-additive peaks near genes with sub-additive combined responses (with the exception of a small increase at high dose; *Figure 3A*). This lack of association suggests that saturation of DNA binding sites may not be sufficient to explain sub-additive combined transcriptional responses; instead, the sub-additive behavior may be a property specifically encoded through the interactions between regulatory factors. It could also be that chromatin accessibility does not quantitatively reflect saturating transcription factor binding.

Our combined transcriptional responses were measured using bulk RNA-sequencing, which averages the transcriptional effects of retinoic acid and TGF-β across millions of cells. Heterogeneity in the response of individual cells could mean that what we observed, for instance, as a multiplicative transcriptional response at the population level is actually a mixture of sub and super-multiplicative transcriptional responses at the single-cell level. Future studies might combine microfluidic delivery of cell signals with live imaging of transcription to measure the response to both individual and combined signal treatments in the same single cells, thereby revealing the extent to which the combined response factor for a given gene displays cell-to-cell heterogeneity (*Zhang et al., 2019*). High amounts of heterogeneity could suggest a need for even greater flexibility in biophysical models of combined transcriptional responses.

In our dataset, the combined response factor remained largely constant over a range of doses. This constancy suggests that whatever the functional interaction is between the factors responsible for the particular mode of combined response, that interaction is quantitatively maintained through doses (with some evidence for saturation at high dose). Such behavior may constrain potential models for interactions, because in principle the interactions could be highly dose dependent. Another open question is whether the mode of combined response for a particular gene depends on the particular signals applied or contextual factors that may vary between cell lines. Further studies may reveal these dependencies.

Another interesting feature of our data was the general lack of strong correspondence between changes in chromatin accessibility and changes in transcriptional output. While we were able to identify some trends, we could not find any strict rules for e.g. what transcription factors associated with what types of combined responses. We found this lack of correspondence surprising, given that transcription factors are the dominant form of transcriptional regulation. There are many potential explanations for this observation. One is that the degree of chromatin accessibility is not as correlated with aggregate transcription factor occupancy levels as we expected. For instance, it may be that accessibility may only change for some types of transcription factor-DNA interactions and not others. Another possibility is that our analysis does not take into account precisely which peaks near a given gene correspond to regulatory elements and which ones do not. This mapping remains largely unknown, although information about what pieces of chromatin spatially contact which other ones may help narrow down the choices (*Fulco et al., 2019*; *Jin et al., 2013*; *Rao et al., 2014*; *Ruf et al., 2011*). Finally, it is also simply possible that the rules governing transcriptional output are highly complex and thus not straightforward to discern from the analyses we performed. In particular, it could be that the genome sequence itself is simply too limited to provide enough sampling of the possible configuration space of transcription-factor-binding motifs to extract rules. The use of

massively parallel reporter assays (*Kwasnieski et al., 2014*; *Patwardhan et al., 2012*) or similar synthetic approaches (*Bogard et al., 2019*; *Rosenberg et al., 2015*) may help reveal such rules.

# Materials and methods

## Key resources table

| Reagent type (species) or resource | Designation | Source or reference | Identifiers | Additional information |
|---|---|---|---|---|
| Cell line (*Homo sapiens*) | MCF-7 (breast carcinoma) | ATCC | ATCC HTB-22, (lot 64125078), RRID:CVCL_0031 | |
| Peptide, recombinant protein | TGF-β | Sigma | Cat# T7039 | |
| Chemical compound, drug | All trans retinoic acid | Sigma | Cat # R2625 | |
| Other | Charcoal- stripped FBS | Gemini | Cat # 100–119 | |
| Commercial assay or kit | miRNeasy RNA extraction kit | Qiagen | Cat # 217004 | |
| Commercial assay or kit | NEBNext Poly(A) mRNA Magnetic Isolation Module | New England Biolabs | Cat # E7490 | |
| Commercial assay or kit | NEBNext Ultra II RNA Library Prep Kit for Illumina | New England Biolabs | Cat # E7770 | |
| Sequence-based reagent | NEBNext Multiplex Oligos for Illumina | New England Biolabs | Cat # E7600 | |
| Commercial assay or kit | Tagment DNA Enzyme and Buffer | Illumina | Cat # 20034197 | |
| Sequence-based reagent | ATAC-seq indices (custom oligos) | Integrated DNA Technologies | | See (*Buenrostro et al., 2013*) for custom index sequences |
| Antibody | Anti-RARA (Rabbit polyclonal) | Sigma | Cat # HPA058282, RRID:AB_2683666 | 1:200 dilution |
| Antibody | Anti- pSMAD2 (Rabbit monoclonal) | Cell Signaling Technology | Cat # 18338T, RRID:AB_2798798 | 1:800 dilution |
| Antibody | Anti-rabbit IgG, Alexa Fluor 647 (Goat polyclonal) | Thermo Fisher Scientific | Cat # A-21244, RRID:AB_2535812 | 1:1000 dilution |

## Cell culture and signal delivery

We acquired one vial of MCF-7 cells from ATCC (lot 64125078), which we expanded in DMEM/F12 with 5% FBS and 1% penicillin/streptomycin. Prior to adding retinoic acid and TGF-β, the cells experienced a total of 13 passages and 1 freeze/thaw cycle. Because normal FBS can have significant amounts of retinoic acid (*Napoli, 1986*), we cultured the cells in a modified medium containing charcoal-stripped FBS, with each batch consisting of 50 ml charcoal-stripped FBS (Gemini, 100–119), 5 ml penicillin/streptomycin (Invitrogen, 15140–122), and 500 ml DMEM/F12 (Gibco, 10565018). We grew the MCF-7 cells in this charcoal-stripped FBS-containing medium for a total of 70 or 71 days prior to treating them with retinoic acid and TGF-β. Our MCF-7 cells were negative for mycoplasma contamination after all RNA and ATAC sequencing experiments, and we validated our MCF-7 cells' identity using ATCC's human STR profiling cell authentication service.

For our dose-response experiment, we split two ~ 80% confluent 10 cm dishes equally into 12 different 10 cm dishes, and waited 24 hr prior to adding media containing retinoic acid (Sigma, R2625), TGF-β (Sigma, T7039), or both signals. Because the cells grew faster when exposed to retinoic acid and slower when exposed to TGF-β, we included two additional control conditions that had 50% and 150% of the starting cell density to test for potential cell-density effects (these additional conditions covered the range of cell-densities seen at the endpoint of our experiments). We treated cells for 72 hr in three doses of retinoic acid (50 nM, 200 nM, and 400 nM), TGF-β (1.25 ng/ml, 5 ng/ml, 10 ng/ml), or both signals (50 nM retinoic acid + 1.25 ng/ml TGF-β, 200 nM RA + 5 ng/ml TGF-β, 400 nM RA + 10 ng/ml TGF-β). The medium dose we chose for TGF-β, 5 ng/ml, is used in several studies of MCF-7 cells (*Mahdi et al., 2015*; *Noman et al., 2017*; *Tian and Schiemann, 2017*), and

the medium dose we used for retinoic acid, 200 nM, is between the 100 nM dose used in *Hua et al., 2009* and the 1 uM dose used in *Cunliffe et al., 2003*. All conditions had the same 0.0125% concentration of ethanol. At 72 hr, we then trypsinized the cells in each well, removing 50,000 of them for immediate ATAC-seq library preparation and lysing the rest of them in Qiazol (storing immediately at −80°C) for subsequent RNA extraction and bulk RNA-seq library preparation.

## Immunofluorescence experiments and imaging

For immunofluorescence experiments, we seeded eight-well glass chambers (Lab-tek 12-565-470) with hormone-starved MCF-7 cells for 24 hr before treating the cells with the medium dose of TGF-β (5 ng/ml), retinoic acid (200 nM), or vehicle (0.0125% ethanol). Following treatment, we fixed cells for 12 min in 3.7% formaldehyde (Sigma F1635) diluted in 1x PBS. We stored samples at 4C in 1x PBS, then performed the immunofluorescence protocol exactly as described by Cell Signaling Technology, using a dilution of 1:800 for the primary anti-pSMAD2 antibody (Cell Signaling Technology 18338T), 1:200 for the primary anti-RARA antibody (Sigma HPA058282), and 1:1000 for the goat anti-rabbit secondary antibody conjugated with Alexa Fluor 647 (Thermo Fisher Scientific A-21244). In brief, we blocked samples with 5% goat serum for 60 min, incubated with primary antibody overnight at 4C, washed three times with 1X PBS for 10 min each, incubated with secondary antibody at room temperature for 90 min in the dark, then washed the cells another three times in 1X PBS. We stained cellular nuclei with DAPI prior to imaging. We imaged the cells with an inverted Nikon TI-E microscope with a 20x Plan-Apo λ (Nikon MRD00205) objective and with DAPI and Atto647N filter sets. We collected all images at 20x magnification.

## Immunofluorescence image analysis

To quantify the nuclear pSMAD2 and RARA signal in our immunofluorescence experiments, we developed a custom image analysis pipeline in Python that was centered around the usage of Cellpose (*Stringer et al., 2020*) to detect the nuclear boundaries of each cell. We first used the DAPI channel to manually select three to six high-quality images per condition. High-quality images had minimal stacking of cells, little correlation between DAPI and immunofluorescence signal, and had well-focused nuclei throughout the image. We then used the DAPI channel images as input to Cellpose, with an expected diameter parameter of 32 pixels. Using Cellpose's identified nuclear boundaries, we then calculated the average intensity inside each nucleus using the corresponding immunofluorescence channel (pSMAD2 or RARA). To correct for differences in background, we then subtracted the average intensity of the annulus surrounding each nucleus in each image, using a disc-shaped structured element, the SciPy binary_dilation function, and the nuclear mask matrix defined by Cellpose to generate the surrounding annulus for each nucleus. We then used this normalized nuclear intensity value for comparing the pSMAD2 and RARA levels between each condition.

## RNA extraction, library preparation, and sequencing

We extracted RNA from previously frozen MCF-7 cell Qiazol lysates using the Qiagen miRNeasy kit (217004). We then used the NEBNext Ultra II RNA Library Prep Kit for Illumina (E7770) with the NEBNext Poly(A) mRNA Magnetic Isolation Module (E7490) and NEBNext Multiplex Oligos for Illumina (E7600) to prepare individual libraries. We then pooled our three replicates' libraries together and performed paired-end sequencing on an Illumina NextSeq 500, using a 75-cycle NextSeq 500/550 High Output Kit v2.5 (20024906), yielding ~15 million read pairs per sample.

## RNA-sequencing analysis pipeline

We aligned reads to the hg38 assembly using STAR v2.7.1a and counted uniquely mapped reads with HTSeq v0.6.1 and the hg38 GTF file from Ensembl (release 90). We performed differential expression analysis using DESeq2 v1.22.2 (*Love et al., 2014*) in R 3.5.1, using a minimum absolute-value log-fold-change of 0.5 and a q value of 0.05. For genes with multiple possible transcription start sites, we used the genomic coordinates of the 'canonical' transcription start site available in the knownCanonical table from GENCODE v29 in the UCSC Table Browser.

## ATAC library preparation and sequencing

At the endpoint of each cell condition, we immediately performed the Omni-ATAC protocol (*Corces et al., 2017*) on 50,000 live MCF-7 cells, using Illumina Tagment DNA Enzyme TDE1 (20034197) at the tagmentation step and double-sided bead purification at the endpoint with Agencourt AMPure XP magnetic beads (A63880). The exact protocol we used is available in the protocols folder at https://github.com/emsanford/combined_responses_paper (*Sanford, 2020a*; copy archived at swh:1:rev:e25f3d9eefd72ac1ab2885d9b0f3ad0c3cf0b3b8). We then performed paired-end sequencing using one 75-cycle NextSeq 500/550 High Output Kit v2.5 (20024906) for each replicate, yielding ~42 million read pairs per sample.

## ATAC-sequencing analysis

We created a paired-end read analysis pipeline using the ENCODE ATAC-seq v1 pipeline specifications. Briefly, we aligned our ATAC-seq reads to the hg38 assembly using bowtie2 v2.3.4.1, filtered out low-quality alignments with samtools v1.1, removed duplicate read pairs with picard 1.96, and generated artificial single-ended text-based alignment files containing inferred Tn5 insertion points with custom Python scripts and bedtools v2.25.0. To call peaks, we used MACS2 2.1.1.20160309 with the command, 'macs2 callpeak –nomodel –nolambda –keep-dup all –call-summits -B –SPMR –format BED -q 0.05 –shift 75 –extsize 150'. While we created this pipeline for use on the Penn Medicine Academic Computing Services' high performance cluster, it is also publicly available at github.com/arjunrajlaboratory/atac-seq_pipeline_paired-end (*Sanford, 2020b*; copy archived at swh:1:rev:c4c819e3ad5828b953fbd2ec05163e590518ae4b). Our pipeline generates a series of post-sequencing quality control metrics, which we have provided in *Supplementary file 1*.

Since we had three biological replicates per ATAC-seq condition, we used an established 'majority rule' to retain only the peak summits that were found in at least two replicates (*Yang et al., 2014*) (we used a peak size of 150 bp, centered on MACS2 summit locations, to mimic the span of one nucleosome). Using these condition-specific peak files, we then used bedtools to create one 'master consensus peak file' by merging each condition's peak summit file together in a manner that disallowed overlapping peaks. We then used the number of ATAC-seq fragment counts at each peak in this master consensus peak file for differential peak analysis.

We wrote a custom peak analysis algorithm that took advantage of our additional ethanol control conditions to estimate a false discovery rate for differential peak identification. In this algorithm, we first count the number of ATAC-seq reads at each peak in the master consensus peak file. We then normalize the fragment counts at each peak to correct for differences in total sequencing depth. In this normalization step, we divide the number of reads in peaks for a given sample by $\frac{sample's\ total\ number\ of\ reads\ in\ peaks}{average\ number\ of\ reads\ in\ peaks\ across\ all\ samples}$. Then, for each condition, we calculate the average number of normalized read counts at each peak. Following this, we fill in an estimated false discovery rate in each cell of a 50 × 50 grid containing 50 exponentially-spaced steps of minimum fold-change values (ranging from 1.1 to 10) and 50 exponentially-spaced steps of minimum number of normalized fragment counts in the condition with the larger number of counts (ranging from 10 to 237). To calculate the estimated false discovery rate, we counted the number of differential peaks between signal-treated conditions and the normal density ethanol control as well as the number of differential peaks between additional ethanol controls (50% and 150% starting cell density) and the normal density ethanol control. We then used the average number of differential peaks in the additional controls to estimate the number of false positive peaks per experimental condition, then calculated the final estimated false discovery rate (FDR) for a given parameter pair using the following formula:

$$estimated\ FDR = \frac{(number\ of\ conditions)(estimated\ number\ of\ false\ positive\ peaks\ per\ condition)}{total\ number\ of\ differential\ peaks\ in\ experimental\ conditions}$$

After calculating the estimated FDR for each cell of the 50 × 50 grid, we then pooled together the differential peaks contained in any cell containing an FDR less than 0.25%. After pooling together the peaks in each of these cells and counting the number of differential peaks in the signal-treated conditions and additional controls, the combined estimated FDR was 0.65%. We then noticed that our original peak set's fixed nucleosomal peak size of 150 bp led to many genomic regions containing several adjacent peaks that appeared to form a single, larger peak. Because of this, we merged our peaks together when they were within 250 base pairs of each other, then we

performed a second round of the same differential peak calling algorithm on the merged peaks, requiring a minimum fold change of 1.5 and a minimum normalized fragment count value of 30. In this final peak set, there are a total of 34,323 differential peaks, with a pooled estimated false discovery rate of 0.43%.

We performed motif analysis on our set of differential peaks using chromVAR v1.5.0 (*Schep et al., 2017*), its associated curated cisBP database of transcription factor motifs, and the motifmatchR Bioconductor package. We treated each replicate as one sample for a given condition, and we pooled together the different dosages of the same signal(s) to decrease the variance of the transcription factor motif deviation scores for retinoic acid, TGF-β, and combined treatment. We slightly modified the chromVAR code to extract an internal metric that equals the fractional change in fragment counts at motif-containing peaks for a given motif.

## Statistical model for categorical classification of combined responses

For a given gene in a given experimental condition, we assumed that its transcripts per million (TPM) value for one replicate was drawn from a Gaussian distribution. We estimated the parameters of these Gaussian distributions to create an 80% confidence interval for which to compare additive and multiplicative predictions. For each dosage of the combination treatment, we classified a gene as sub-additive if the additive and multiplicative predictions were higher than the 80% confidence interval, additive if only the additive prediction laid in the confidence interval, multiplicative if only the multiplicative prediction laid in the interval, super-multiplicative if both additive and multiplicative predictions were below the confidence interval, and ambiguous if both the additive and the multiplicative prediction laid within the interval.

To estimate the mean expression value of a gene in an experimental condition (e.g. 200 nM retinoic acid), we simply calculated the average TPM value across the three replicates. To improve our variance estimates, we took advantage of an observation we made during extensive manual review that the coefficient of variation (CV) appeared to be the same between each dosage we tested for retinoic acid, TGF-β, and combined treatment (*Figure 1—figure supplement 1E–F*). We then assumed that each dosage of a condition shared one CV term, which we calculated by averaging each dose's CV estimate using the unbiased estimator:

$$CV(gene,\ signal,\ dosage) = \left(1 + \frac{1}{4n}\right)\frac{s}{\bar{x}}$$

$$CV(gene,\ signal) = \frac{1}{m}\sum_{1}^{m}CV(gene,\ signal,\ dosage_m)$$

where $n$ is the number of replicates (three in our case), $s$ is the sample standard deviation, and $\bar{x}$ is the mean of the measured TPM values, and m is the number of doses tested (three in our case). Finally, we used this averaged CV estimate to estimate a variance parameter for the Gaussian distribution we assumed to underlie the TPM values for a given gene and signal. For a given gene, dosage, and signal, our final estimated Gaussian distribution was:

$$TPM(gene,\ signal,\ dosage) \sim Gaussian\left(\overline{x_{gene,signal,dosage}}\ ,\ \left(\overline{x_{gene,signal,dosage}} \times CV(gene,\ signal)\right)^2\right)$$

Where $\overline{x_{gene,signal,dosage}}$ is the measured average TPM value for a given gene exposed to a specific dose of retinoic acid, TGF-β, or combination treatment. The benefit of using our shared CV term across dosages was to move from using the information from three samples to using the information from nine samples when estimating the variances of these distributions.

To classify ATAC-seq peaks as sub-additive, additive, or super-additive, we used the same approach described above for RNA-seq TPM values, but with a given peak's normalized fragment count value. We then classified peaks as sub-additive or super-additive if the additive prediction was higher than or lower than (respectively) the estimated Gaussian distribution's 80% confidence interval.

## Statistical model for simulated additive and multiplicative predictions

To simulate new ATAC-seq and RNA-seq measurements, for each gene and condition we randomly sampled three new observations from a folded Gaussian distribution (folded to avoid negative expression or normalized fragment count values) with the parameters we previously estimated for the purpose of categorically classifying combined response behaviors. For the combined treatment, we set the mean of the distribution to be either a perfectly additive or perfectly multiplicative prediction. We then calculated the average of the three new simulated observations and used these average values to determine a gene's c value at a given dose or an ATAC-seq peak's d-value at a given dose. Using this process, we calculated 250 simulated c values for each dose of each upregulated gene in our master set and 10 simulated d values for each ATAC-seq peak that was upregulated individually by retinoic acid and TGF-β. In the simulated data mixture model where genes can be strictly additive or multiplicative, at each we randomly assigned a gene to be additive or multiplicative based on the ratio of the dose-specific frequencies we observed in the categorical classification of the combined response.

## Use of simulated data to infer the location of a secondary peak in the observed combined response factor (c value) histogram

To generate a hypothetical plot of observed c values in which the primary peak of additive responses centered at c = 0 was depleted, we subtracted the additive component of a c value histogram generated by simulated data. These simulated c values were generated using gene and condition-specific Gaussian distributions in a process outlined above and in *Figure 1—figure supplement 1*. At each dose, we simulated data as a mixture of additive and multiplicative combined responses, setting the exact proportion of simulated additive versus multiplicative combined responses based on the ratio of additive to multiplicative combined transcriptional responses seen at each dose of the observed data (*Figure 1E*; *Figure 1—figure supplement 1B*). We then scaled the size of this 'mixed' simulated c value distribution to the peak heights at c = 0 and c = 1 in the observed c value histogram by minimizing the squared distance between the simulated and observed histogram bars directly abutting c = 0 and c = 1 (4 histogram bars total). We then subtracted the additive component of the simulated c value distribution and locally (in the range of c = −4 to c = 5) fit a Gaussian density function to the residual histogram using the nls function in R.

We also estimated the probability of obtaining the number of combined transcriptional responses in each bin of our observed c value histogram if all combined responses were additive. To do this, we scaled the peak height of our observed data at c = 0 to the peak height of an additively simulated distribution of c values. We then repeatedly (1000 times) ran new simulations of additive combined responses, simulating one observation per gene in our master set of 1384 genes. We used a bin width of 0.25 and allowed for overlapping bins. Because the probability of obtaining the observed number of counts was extremely low for many bins and because the variability in the number of observations in a given bin was well described by a Poisson distribution (outside the range of −0.3 < c < 0.3), we used a Poisson cumulative density function to estimate the probability of witnessing the number of observed counts (or greater) in each c value bin of the simulated additive data.

## Generating a null distribution for dual-motif matches

To generate a null distribution for dual-motif matches, we first separated our set of upregulated peaks into sub-additive, additive, and super-additive peaks. Within these peak subsets, we counted the number of retinoic acid-dominant (FOX, and ETS-family factors), TGF-β-dominant (SMAD, AP-1, BACH, BATF, SMARCC1, NFE2, NFE2L2, MAFF, and MAFK), and neither-signal-dominant (HOX, NFKB, CDX, CTCF, BCL, and GRHL1) motifs at each peak. Due to similar features of their position-weight matrices, we avoided over-counting similar motifs by reporting the maximum number of motif matches for a single type of motif within a group of motifs. The motif groups we used were as follows: retinoic acid receptor consisted of RARA, group FOX consisted of FOXA1, FOXA2, FOXA3, FOXC2, FOXD3; group ETS consisted of SPI, SPIB, SPIC, EHF, ELF1, ELF2, ELF3, ELF4, ELF5; group SMAD consisted of SMAD3, SMAD4, SMAD9; group AP-1 consisted of JUN, JUNB, JUND, JDP2, FOS, FOSB, FOSL1, FOSL2, BACH1, BACH2, BATF (note the inclusion of non-canonical AP-1 factors due to their similar motif position weight matrices); group SMARCC1 consisted of SMARCC1; group

NFE consisted of NFE2, NFE2L2; group MAF consisted of MAFF, MAFK; group HOX consisted of HOXA13, HOXB13, HOXC10, HOXC12, HOXC13, HOXD13; group NFKB consisted of NFKB1, REL, RELA; group CDX consisted of CDX1, CDX2; group CTCF consisted of CTCF; group BCL consisted of BCL11A, BCL11B; group GRHL1 consisted of GRHL1. For example, if a peak had three JUN motifs, two FOS motifs, two JDP2 motifs, and one BACH1 motif, we would count this as three AP-1 motifs. We then randomly shuffled these grouped motif matches within each peak set, with each peak retaining its original number of total motif matches (thus a peak with zero motif matches also had zero motif matches and a peak with four grouped motif matches always had four grouped motif matches after each random shuffle). After each of 1000 random shuffles, we calculated the fraction of peaks in each peak set that contained both a retinoic acid-dominant and a TGF-β dominant motif.

## Statistical analysis

With the exception of DeSeq2's adjusted p value and our manually calculated p value for the null distribution we generated for dual-motif matches at upregulated ATAC-seq peaks, we calculated all reported p values in the figures and main text using Welch's unequal variances t-test in R. (Note that we did not correct for multiple comparisons.)

## Data and code availability

All custom data analysis code is available at https://github.com/emsanford/combined_responses_paper. The ATAC-seq pipeline we used is available at https://github.com/arjunrajlaboratory/atac-seq_pipeline_paired-end (copy archived at swh:1:rev:c4c819e3ad5828b953fbd2ec05163e590518ae4b). The RNA-seq pipeline we used is available at https://github.com/arjunrajlaboratory/RajLabSeqTools (*Sanford et al., 2020*; copy archived at swh:1:rev:c8b8c79b2ec9c1bd9eb7ced427bb2aec25f19506).

## Acknowledgements

We thank John Murray, Ken Zaret, Golnaz Vahedi, Nancy Zhang, Caroline Bartman, and Yogesh Goyal for helpful ideas and conversations, Connie Jiang, Chris Coté and Ryan Boe for assistance with figure design, Connie Jiang for assistance with cell culture, Amanpreet Kaur, Naveen Jain, and Ian Dardani for assistance with immunofluorescence and associated imaging, Lauren Beck and Phil Burnham for assistance with immunofluorescence image analysis, and Caroline Bartman and Yogesh Goyal for critical review of the manuscript. EMS acknowledges support from NIH training grant F30 HG010986, BLE acknowledges support from NIH training grants F30 CA236129, T32 GM007170 and T32 HG000046, AJC acknowledges support from NIH training grant T32 GM-07229, and AR acknowledges support from R01 CA238237, NIH Director's Transformative Research Award R01 GM137425, R01 CA232256, NSF CAREER 1350601, P30 CA016520, SPORE P50 CA174523, NIH U01 CA227550, NIH 4DN U01 HL129998, NIH Center for Photogenomics (RM1 HG007743), and the Tara Miller Foundation.

# Additional information

### Competing interests

Arjun Raj: Receives consulting income and royalties related to Stellaris RNA FISH probes. The other authors declare that no competing interests exist.

### Funding

| Funder | Grant reference number | Author |
| --- | --- | --- |
| National Institutes of Health | R01 CA238237 | Arjun Raj |
| National Institutes of Health | Transformative Research Award R01 GM137425 | Arjun Raj |
| National Institutes of Health | R01 CA232256 | Arjun Raj |
| National Science Foundation | CAREER 1350601 | Arjun Raj |

| National Institutes of Health | U01 CA227550 | Arjun Raj |
| National Institutes of Health | U01 HL129998 | Arjun Raj |
| National Institutes of Health | RM1 HG007743 | Arjun Raj |
| National Institutes of Health | P30 CA016520 | Arjun Raj |
| National Institutes of Health | F30 CA236129 | Benjamin L Emert |
| National Institutes of Health | T32 GM007170 | Benjamin L Emert |
| National Institutes of Health | T32 HG000046 | Benjamin L Emert |
| National Institutes of Health | T32 GM- 07229 | Allison Coté |
| Tara Miller Foundation | | Arjun Raj |
| National Institutes of Health | SPORE P50 CA174523 | Arjun Raj |
| National Institutes of Health | F30 HG010986 | Eric M Sanford |

The funders had no role in study design, data collection and interpretation, or the decision to submit the work for publication.

## Author contributions

Eric M Sanford, Conceptualization, Data curation, Software, Formal analysis, Investigation, Visualization, Methodology, Writing - original draft, Writing - review and editing; Benjamin L Emert, Supervision, Methodology, Writing - review and editing; Allison Coté, Supervision, Writing - review and editing; Arjun Raj, Conceptualization, Resources, Supervision, Funding acquisition, Investigation, Methodology, Writing - original draft, Project administration, Writing - review and editing

## Author ORCIDs

Eric M Sanford (iD) https://orcid.org/0000-0002-9232-9334
Arjun Raj (iD) https://orcid.org/0000-0002-2915-6960

## Decision letter and Author response

Decision letter https://doi.org/10.7554/eLife.59388.sa1
Author response https://doi.org/10.7554/eLife.59388.sa2

# Additional files

## Supplementary files

- Supplementary file 1. Post-sequencing ATAC-seq metrics for each sample.

- Transparent reporting form

## Data availability

We have uploaded our sequencing results to NIH GEO. Analysis Code: https://github.com/emsanford/combined_responses_paper; https://github.com/arjunrajlaboratory/atac-seq_pipeline_paired-end; https://github.com/arjunrajlaboratory/RajLabSeqTools (copy archived at https://archive.softwareheritage.org/swh:1:rev:e25f3d9eefd72ac1ab2885d9b0f3ad0c3cf0b3b8/).

The following dataset was generated:

| Author(s) | Year | Dataset title | Dataset URL | Database and Identifier |
| --- | --- | --- | --- | --- |
| Sanford EM, Emert BL, Cote A, Raj A | 2020 | Gene regulation gravitates towards either addition or multiplication when combining the effects of two signals | https://www.ncbi.nlm.nih.gov/geo/query/acc.cgi?acc=GSE152749 | NCBI Gene Expression Omnibus, GSE152749 |

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

## Appendix 1

### Mathematical formulation of an additive combined response, multiplicative combined response, and the combined response factor (c value)

Suppose that gene *X* is expressed at baseline and increases its transcription in response to either signal A or signal B:

$$\text{expression of gene X at baseline} = X_{\text{baseline}}$$
$$\text{expression of gene X after receiving signal A} = X_{\text{baseline}} + \Delta_A$$
$$\text{expression of gene X after receiving signal B} = X_{\text{baseline}} + \Delta_B$$

If the combined transcriptional response to receiving both signals A and B were additive, the increase in transcription of gene X would reflect the sum of the effects $\Delta_A$ and $\Delta_B$:

$$\text{additive combined response of gene X to signals A and B} = X_{\text{baseline}} + \Delta_A + \Delta_B$$

If the combined response were multiplicative, the increase in transcription of gene *X* would reflect the product of the fold change experienced under signals A and B:

$$\begin{aligned}
\text{multiplicative combined response} \ &= X_{\text{baseline}} \times \text{fold-change}_A \times \text{fold-change}_B \\
&= X_{\text{baseline}} \times \frac{X_{\text{baseline}} + \Delta_A}{X_{\text{baseline}}} \times \frac{X_{\text{baseline}} + \Delta_B}{X_{\text{baseline}}}
\end{aligned}$$

Multiplying out the terms of the previous expression, we see that the difference between an multiplicative and additive combined response is exactly $\frac{\Delta_A \times \Delta_B}{X_{\text{baseline}}}$:

$$\begin{aligned}
\text{multiplicative response} \ &= X_{\text{baseline}} \times \frac{X_{\text{baseline}} + \Delta_A}{X_{\text{baseline}}} \times \frac{X_{\text{baseline}} + \Delta_B}{X_{\text{baseline}}} \\
&= X_{\text{baseline}} \times \frac{X_{\text{baseline}}^2 + (X_{\text{baseline}} \times \Delta_A) + (X_{\text{baseline}} \times \Delta_B) + (\Delta_A \times \Delta_B)}{X_{\text{baseline}}^2} \\
&= \frac{X_{\text{baseline}}^2 + (X_{\text{baseline}} \times \Delta_A) + (X_{\text{baseline}} \times \Delta_B) + (\Delta_A \times \Delta_B)}{X_{\text{baseline}}} \\
&= X_{\text{baseline}} + \Delta_A + \Delta_B + \frac{\Delta_A \times \Delta_B}{X_{\text{baseline}}} \\
&= \text{additive response} + \frac{\Delta_A \times \Delta_B}{X_{\text{baseline}}}
\end{aligned}$$

We defined a term, *c*, the combined response factor, that can be determined after measuring a gene's expression at baseline and in response to both single and combined signal treatments:

$$\text{gene X's combined response} = X_{\text{baseline}} + \Delta_A + \Delta_B + c \times \frac{\Delta_A \times \Delta_B}{X_{\text{baseline}}}$$

For a gene that increases transcription in response to both signals, the combined response is perfectly additive when $c = 0$, perfectly multiplicative when $c = 1$, sub-additive when $c < 0$, and super-multiplicative when $c > 1$. Thus, a gene's combined response factor, which can be solved for after profiling gene expression in unperturbed and signal-treated cells, provides us with a metric for describing combined transcriptional responses along a continuum that spans addition and multiplication.

