## [Decision Letter]

**Acceptance summary:**

The work investigates an interesting question in gene regulation: when two independent transcription induction signals on one gene are combined, will it result in an additive or multiplicative gene expression level? Using genomic approaches to assay both mRNA expression and chromatin accessibility, the work finds that while there is a large range of responses, most genes favor either additive or multiplicative outcome. This work connects the phenomenological description of gene expression with a mechanistic insight of how two independent transcription signals could work together to regulate one gene.

**Decision letter after peer review:**

Thank you for submitting your article "Gene regulation gravitates towards either addition or multiplication when combining the effects of two signals" for consideration by *eLife*. Your article has been reviewed by two peer reviewers, and the evaluation has been overseen by a Reviewing Editor and Naama Barkai as the Senior Editor. The following individuals involved in review of your submission have agreed to reveal their identity: Hernan Garcia (Reviewer #1); Angela H DePace (Reviewer #2).

The reviewers have discussed the reviews with one another and the Reviewing Editor has drafted this decision to help you prepare a revised submission.

We would like to draw your attention to changes in our revision policy that we have made in response to COVID-19 (https://elifesciences.org/articles/57162). Specifically, we are asking editors to accept without delay manuscripts, like yours, that they judge can stand as *eLife* papers without substantial additional data, even if they feel that they would make the manuscript stronger. Thus the revisions requested below mainly address clarity and presentation, with a few minor experiments if possible.

Summary:

In this work the authors used a genomic approach to investigate the way cells interpret two combined signals versus two individual signals. The authors used RNA-seq to examine the gene expression outputs from thousands of genes in response to two signal inputs, TGF-β and retinoic acid, either individually or in combination. The authors found that when stimulated with both signals, most cells exhibited additive or multiplicative responses. The authors further used paired chromatin accessibility by ATAC-seq to relate such responses to putative transcription factory binding patterns in these genes. Surprisingly, ATAC-seq revealed that most genes prefer addition to combine two signals as chromatin accessibility is largely additive, although some super-additive accessibility may respond to multiplicative gene expression.

This work provides a platform to quantitatively assess combinatorial transcription regulation both at the level of DNA accessibility and transcriptional output. Although the concept of additive vs. multiplicative transcriptional response is phenomenological, it may be used to clarify and constrain certain biophysical models of transcriptional regulation and set the stage for a better understanding of the molecular relation between combinatorial transcription factor binding and corresponding gene activity.

While the work is written in a clear and concise language, there are places that require further clarification and better presentations.

Essential revisions:

1) Presentations of the logic and data: it is hard to follow the math in the absence of equations:

a) Please show the equations featured in Figure 1A in the main text together with some derivation or explanation that can build intuition.

b) The simulations results shown in Figure 1 and Figure 2 are a key part of the argument. Yet, their details are buried in the SI, making it hard to follow their justification (which could also benefit from schematics). Please explain these simulations in the context of the main text.

c) "For each dosage of the combination treatment, we classified a gene as sub-additive if the additive and multiplicative predictions were higher than the 80% confidence interval, additive if only the additive prediction laid in the confidence interval, multiplicative if only the multiplicative prediction laid in the interval, super-multiplicative if both additive and multiplicative predictions were below the confidence interval, and ambiguous if both the additive and the multiplicative prediction laid within the interval." Please show this graphically and/or with an equation.

2) Clarifications of experimental conditions and results:

a) The authors treat the cells for 72h. This is a very long time where secondary effects may be dominating the results. The choice of this time point should, at the very least, be justified and discussed. For example, previous studies that quantitatively characterized distinct temporal dynamics in SMAD signaling after TGF-β treatment showed a transient, dose dependent SMAD response in the first 4 h after TGF-β treatment, with a strong early peak in the nuclear/cytoplasmic ratio of SMAD2/4 (Clarke and Liu, 2008; Schmierer et al., 2008; Zi et al., 2011; Zi et al., 2012; Strasen et al., 2018). In addition, TGF-β signaling has been suggested to depend on cell density and cell cycle stage (Zieba et al., 2012), which may also affect the results. Also it would be helpful to have a quantitative measure of the corresponding nuclear TF levels at the selected time-point after 72h (e.g. for main affected TFs such as pSMAD2 and RARA levels).

b) MCF7 cells were treated with three different doses of TGF-β (1.25, 5, and 10 ng/mL) and RA (50, 200 and 400 nM). As it seems that the selected doses are higher than what has been used in previous studies, the authors should comment on their choice.

c) The authors state that "We defined a master set of 1,398 upregulated genes by selecting the set of genes that were differentially expressed in any dose of the combination treatment (log FC {greater than or equal to} 0.5 and padj {less than or equal to} 0.05) and that had increased expression in each dose of each individual signal." It is unclear how this gene set relates to the top-right Venn diagram in Figure 1B, in which only 303 genes are shown as being upregulated in all three treatments and the total according to the numbers in the diagram are >1398.

d) Figure 1B shows that a large proportion of genes were differentially expressed in response to both signals, but not to either of the signals individually. Their responses are presumably more non-additive than the responses of genes upregulated in response to all three treatments. Restricting analysis to the latter group therefore introduces a bias for certain modes of combinatorial regulation. The justification for this choice should be discussed.

e) The authors frame the work on the basis of simple models of gene regulation by pairs of transcription factors that predict either addition or multiplication. However, they are activating two signaling pathways that could interact also at the level of signal transduction (and need not be directly regulating the genes in question, as noted in point 1). How justifiable is it to make inferences about the nature of combinatorial transcriptional regulation from this kind of experimental set up? These issues should be made clearer from the beginning and should be taken into account when interpreting the data.

f) Related to the point above, the authors use chromatin accessibility as a proxy for TF binding. However, this does not need to be the case, especially if the accessibility data are considered quantitatively. For example, TFs may bind and recruit remodeling factors that affect accessibility differentially across the genome, obscuring the relationship between TF binding and accessibility. This is especially pertinent at longer time scales after perturbation. We suggest presenting the data on accessibility as just that, instead of presenting it as data that directly reports on TF binding. The relationship to TF binding can and should still be explored in the analyses, but with clarification for how accessibility data is limited in this case. The following are instances where accessibility data is described as directly reporting on TF binding that we recommend revising (the list is not exhaustive):

– the title of section two

– Figure 2E

– the link between models of TF control and the relationship between peaks and expression, such as the reference to the thermodynamic model at the end of section 3.

– remove the implicit assumption between cooperativity of TF binding and super-additive peaks in section 3 and section 4. This may help explain more naturally the lack of dual-motif finding in section 4.

3) Title: The authors suggest a bimodal distribution for the observed c values, with peaks at 0 and 1. The authors write that "Our simulated c value distributions bear a moderate resemblance to our observed c value distributions". This conclusion is central to the paper's claim that "Gene regulation gravitates towards either addition or multiplication when combining the effects of two signals" (title) and that "the combined responses exhibited a range of behaviors, but clearly favored both additive and multiplicative combined transcriptional responses" (Abstract). However, the additional peak at c=1 is not obvious from the data in Figure 1E. Stronger evidence (i.e. statistical analysis of the observed distributions) would be needed to demonstrate overrepresentation of c values ~1. Alternatively, the title and Abstract could be revised to better reflect the strength of the findings.

---

## [Author Response]

Essential revisions:1) Presentations of the logic and data: it is hard to follow the math in the absence of equations:a) Please show the equations featured in Figure 1A in the main text together with some derivation or explanation that can build intuition.

We thank the reviewers for pointing out this lack of clarity and intuition in the presentation of our equations in Figure 1A. To address this issue, we created an appendix referenced early in the main text, which performs a step-by-step derivation of the additive combined response, multiplicative combined response, the precise difference term between the additive and multiplicative combined responses, and the combined response factor (c value). Please see “Appendix 1: mathematical description of additive versus multiplicative combined transcriptional responses”

b) The simulations results shown in Figure 1 and Figure 2 are a key part of the argument. Yet, their details are buried in the SI, making it hard to follow their justification (which could also benefit from schematics). Please explain these simulations in the context of the main text.

The reviewers are right that the simulations are critical and lend substance to our arguments. We have now described them in further detail in the main text and have created explanatory schematics (Figure 1—figure supplement 1C-D) to help flesh out the motivation and methodology for our analysis and conclusions. We have also performed a far more rigorous analysis of the “c-value” histogram; see point #3 below for further discussion of that point. See below for our updated main text describing how we performed these simulations:

“At all doses of combination treatment, we observed a wide peak centered around c=0 (additive), with a hint of a secondary peak at c=1 (multiplicative), suggesting that the integration of the effects of two signals is preferentially additive or multiplicative (Figure 1F; Figure 1—figure supplement 4). […] The resultant residual distribution was a broad peak centered roughly around c=1 (a Gaussian fit to the residual gave a fit centered at c=1.12 and c=1.00 at medium and high doses, respectively), consistent with our multiplicative simulated data (Figure 1F; Figure 1—figure supplement 4A).”

c) "For each dosage of the combination treatment, we classified a gene as sub-additive if the additive and multiplicative predictions were higher than the 80% confidence interval, additive if only the additive prediction laid in the confidence interval, multiplicative if only the multiplicative prediction laid in the interval, super-multiplicative if both additive and multiplicative predictions were below the confidence interval, and ambiguous if both the additive and the multiplicative prediction laid within the interval." Please show this graphically and/or with an equation.

The reviewers are correct that our original presentation did not do a good job of illustrating our statistical classification scheme. We have now incorporated a supplementary figure (Figure 1—figure supplement 1) with a schematic that we hope illustrates the point better, which we now reference in the main text. We thank the reviewer for suggesting this clarification.

We have also updated the main text to reduce the confusion caused by the very long sentence quoted by the reviewers in this point. The updated text is now as follows:

“We classified a combined transcriptional response as sub-additive, additive, multiplicative, or super-multiplicative by comparing where a “perfect” hypothetical additive or multiplicative response lay with respect to the 80% confidence interval of the combined treatment's expression value (Figure 1—figure supplement 1B). If both the hypothetical additive and the hypothetical multiplicative predictions lay within the confidence interval, we classified the response as ambiguous (Figure 1—figure supplement 1B).”

We thank the reviewers for directing our attention to this point of confusion and hope that our new schematics and changes to the main text add clarity to how we classified genes as sub-additive, additive, multiplicative, super-multiplicative, or ambiguous.

2) Clarifications of experimental conditions and results:a) The authors treat the cells for 72h. This is a very long time where secondary effects may be dominating the results. The choice of this time point should, at the very least, be justified and discussed. For example, previous studies that quantitatively characterized distinct temporal dynamics in SMAD signaling after TGF-β treatment showed a transient, dose dependent SMAD response in the first 4 h after TGF-β treatment, with a strong early peak in the nuclear/cytoplasmic ratio of SMAD2/4 (Clarke and Liu, 2008; Schmierer et al., 2008; Zi et al., 2011; Zi et al., 2012; Strasen et al., 2018). In addition, TGF-b signaling has been suggested to depend on cell density and cell cycle stage (Zieba et al., 2012), which may also affect the results. Also it would be helpful to have a quantitative measure of the corresponding nuclear TF levels at the selected time-point after 72h (e.g. for main affected TFs such as pSMAD2 and RARA levels).

The reviewers have brought up a critical point about the timescales we used in our experiments. It is true that there are likely secondary effects at play, however, in principle such secondary effects do not per se affect the conclusions we draw. Our goal was not to elucidate the precise mechanism by which genes are regulated, but rather to ascertain how they responded to combinations of regulatory signals, regardless of the precise origin of those signals. Indeed, we chose a longer timescale so as to have more genes displaying differential expression, thus enabling us to add more genes to our analysis. It is true that one situation in which secondary effects could muddy our interpretations would be if the signals were largely overlapping (i.e., two signals that activated the retinoic acid pathway), but our chromVAR analysis of our paired ATAC-seq data (Figure 4) showed that the changes in gene expression we measured were associated with largely distinct sets of transcription factors. We apologize for not making this rationale clear in our original manuscript, and we have now added discussion of this point to the main text.

Regarding cell density and cell cycle stage, it is true that these factors could affect the signaling dynamics, and we did not test these factors would have altered our results. We have added some text discussing this caveat in the main text:

“Note that our ethanol “vehicle” controls were performed at three different cell concentrations, and there were no significantly differentially expressed genes between concentrations. We did not, however, add the signals to different concentrations of cells or cells at different points in the cell cycle, in which context the signals may exert differential effects.”

The reviewers have also brought up the specific dynamics of the TGF-β and retinoic acid signaling pathways. We performed a series of immunofluorescence experiments aimed at showing the extent to which the signals were independent and their timescales. We found that in response to TGF-β, nuclear pSMAD2 levels (i.e., activation) increased rapidly (by 40 minutes) and continued to stay above baseline at 72 hours (Figure 1—figure supplement 2); however, retinoic acid had no effect on these levels, demonstrating that there was little-to-no cross-activation of SMADs by TGF-β (Figure 1—figure supplement 2E). (In the other direction, we confirmed that the retinoic acid receptor was essentially always nuclear regardless of activation status, and so it was impossible to discern any signal cross-activation in that case (Figure 1—figure supplement 3); however, we saw no evidence of activation of RARA by TGF-β in our chromVAR analysis (Figure 4A).) In total, we believe these results suggest that the signaling pathways activated by our signals act through largely independent mechanisms, making it possible to interpret our results despite the presence of secondary effects.

We added a new paragraph to our main text to summarize the results of our immunofluorescence experiments and ATAC-seq analysis:

“In our analysis of combined transcriptional responses, we assumed that retinoic acid and TGF-β exhibited their effects on common target genes through distinct transcription factors. […] This same motif analysis also suggested that retinoic acid and TGF-β largely increased the activity of distinct transcription factors at the 72 hour time point, meaning that the secondary effects of retinoic acid and TGF-β are likely mediated through the activity of distinct transcription factors.”

b) MCF7 cells were treated with three different doses of TGF-β (1.25, 5, and 10 ng/mL) and RA (50, 200 and 400 nM). As it seems that the selected doses are higher than what has been used in previous studies, the authors should comment on their choice.

The reviewers have raised an important point about the doses that we used. Most other studies we have found using TGF-β in MCF-7 cells used a concentration of 5 ng/mL (Mahdi et al., 2015; Noman et al., 2017; Tian and Schiemann, 2017), which was our “medium” dose used, and which gave a robust transcriptional response in our hands. Other studies using retinoic acid in MCF-7 cells used concentrations higher than ours, on the order of 1µM (Cunliffe et al., 2003), but we found that concentration to be lethal in combination with TGF-β, so after some empirical testing, we settled on the range of concentrations we used because they elicited a robust and dose-dependent transcriptional response. We have now commented on the choice of doses and model system in the main text.

c) The authors state that "We defined a master set of 1,398 upregulated genes by selecting the set of genes that were differentially expressed in any dose of the combination treatment (log FC {greater than or equal to} 0.5 and padj {less than or equal to} 0.05) and that had increased expression in each dose of each individual signal." It is unclear how this gene set relates to the top-right Venn diagram in Figure 1B, in which only 303 genes are shown as being upregulated in all three treatments and the total according to the numbers in the diagram are >1398.

We thank the reviewers for pointing out our lack of clarity in describing the gene sets we used. Indeed, the full set of genes in the Venn diagram totalled to 2,246 genes, which included any gene that was upregulated in the combined treatment. However, for the reasons we outline below, we had a further requirement that the transcriptional changes from both individual signals be > 0 in order to be readily interpretable in the additive/multiplicative framework; these restrictions narrowed the gene set to roughly 1,398 genes, as described in our updated text below (and also as marked in an updated figure):

“We defined a master set of 1,398 genes by selecting the set of genes that were significantly upregulated in any dose of the combination treatment (log2 fold-change ≥ 0.5 and Benjamini-Hochberg adjusted p value ≤ 0.05) and that had increased expression in all doses of each individual signal (Figure 1D). […] (There were only two genes that were significantly downregulated in the combined treatment while also having ΔA > 0 and ΔB > 0 at all doses of each individual signal treatment; we elected to also include these two genes in our master set for the total of 1398.)”

We also added a new panel to Figure 1 (Figure 1D) to graphically illustrate this set using an annotated version of the Venn diagram from the top-right of Figure 1B.

Note that in our analyses of the c values of the combined responses in this master set, we removed an additional 14 genes from this set that had an expression of exactly 0 TPM in the control condition. We removed these genes because when the control TPM is zero, the estimation of the c value also becomes exactly zero, implying a perfectly additive combined response when such a response may not be the case (see equations in Figure 1—figure supplement 5A). We have added asterisks with appropriate explanatory footnotes in these cases to the figures and the main text. We hope these changes help clarify the gene sets we used.

d) Figure 1B shows that a large proportion of genes were differentially expressed in response to both signals, but not to either of the signals individually. Their responses are presumably more non-additive than the responses of genes upregulated in response to all three treatments. Restricting analysis to the latter group therefore introduces a bias for certain modes of combinatorial regulation. The justification for this choice should be discussed.

We apologize for the confusion regarding the definition of our master set of upregulated genes, which we now go over in detail as described in the previous point (point 2c above). The reviewers are right that genes significantly upregulated in response to both signals but not in response to either of the signals individually should be included in the analysis, and we did in fact include most, but not all, of these genes (we included 493 of the 752 genes (66%) significantly upregulated in both signals but not either signal individually). The reason we excluded some of these genes relates to our analysis framework, in which we require the effects due to the individual signals to be greater than zero (but not necessarily statistically significantly upregulated) or order to develop an analysis framework with a consistent mapping between our c values and categories of combined responses (e.g. “sub-additive”, “super-multiplicative). We now include a discussion of the selection of the master set and how it relates to the Venn diagram in Figure 1B in the main text, shown below:

“If we had selected the full set of all genes upregulated in any dose of the combined treatment, we would have analyzed a set of 2246 genes (Figure 1D). […] Inclusion of genes with negative changes after individual signal treatments would require a more elaborate analysis framework to encompass the much larger variety of categorizations of potential responses that would be difficult to characterize with the number of genes in our analysis.”

e) The authors frame the work on the basis of simple models of gene regulation by pairs of transcription factors that predict either addition or multiplication. However, they are activating two signaling pathways that could interact also at the level of signal transduction (and need not be directly regulating the genes in question, as noted in point 1). How justifiable is it to make inferences about the nature of combinatorial transcriptional regulation from this kind of experimental set up? These issues should be made clearer from the beginning and should be taken into account when interpreting the data.

The reviewers have made a great point about the potential for cross-interaction. Indeed, this comment motivated us to perform the cross-interactivity immunofluorescence experiments we described in our response to point 2a above. Our interpretation, based on those data in addition to the ATAC-seq analyses, is that it is likely that there is not appreciable cross-activity in the signaling pathways or their downstream effectors. We have made this point also in the main text as per our response to 2a, as well as including appropriate caveats. We appreciate the reviewers raising these concerns, as addressing them has enhanced the manuscript.

f) Related to the point above, the authors use chromatin accessibility as a proxy for TF binding. However, this does not need to be the case, especially if the accessibility data are considered quantitatively. For example, TFs may bind and recruit remodeling factors that affect accessibility differentially across the genome, obscuring the relationship between TF binding and accessibility. This is especially pertinent at longer time scales after perturbation. We suggest presenting the data on accessibility as just that, instead of presenting it as data that directly reports on TF binding. The relationship to TF binding can and should still be explored in the analyses, but with clarification for how accessibility data is limited in this case. The following are instances where accessibility data is described as directly reporting on TF binding that we recommend revising (the list is not exhaustive):– the title of section two– Figure 2E– the link between models of TF control and the relationship between peaks and expression, such as the reference to the thermodynamic model at the end of section 3.– remove the implicit assumption between cooperativity of TF binding and super-additive peaks in section 3 and section 4. This may help explain more naturally the lack of dual-motif finding in section 4.

The reviewers are totally right that the extent to which there is a quantitative relationship between transcription factor occupancy and chromatin accessibility is still unclear, and that our manuscript was too heavily implying that changes in chromatin accessibility are a direct readout of aggregate changes in transcription factor occupancy. We have reviewed and revised the portions of the manuscript (which includes the four instances mentioned by the reviewers as well as the Abstract), presenting the combined responses of ATAC-seq peaks more precisely as changes in chromatin accessibility and tempering any implications of a direct connection between ATAC-seq changes and transcription factor binding. We have included two examples of more significant edits to the manuscript that we hope address the reviewers’ concerns:

“We performed ATAC-seq on the same populations described earlier, reasoning that the observation that changes in chromatin accessibility have been shown to correlate with changes in aggregate transcription factor binding activity (Thurman et al., 2012) meant that we could infer something about transcription factor binding at these sites. […] Reassuringly, our initial motif enrichment analysis revealed that retinoic acid receptor alpha (RARA) and three TGF-β pathway transcription factor motifs (SMAD3, SMAD4, and SMAD9) were highly enriched in their respective individual signal treatment conditions (Figure 4—figure supplement 1B).”

“Super-multiplicative combined responses may reflect cooperative interactions between retinoic acid and TGF-β induced factors, in which binding of a retinoic acid factor to DNA strengthens the binding of a TGF-β factor to nearby DNA or vice versa. […] However, given that ATAC-seq peaks likely have additional routes to super-additive increases in accessibility (perhaps involving chromatin remodeling factors affected by our signals), further work would be needed to demonstrate that super-multiplicative transcriptional responses are indeed a result of direct binding interactions at enhancers between each signal’s induced transcription factors.”

3) Title: The authors suggest a bimodal distribution for the observed c values, with peaks at 0 and 1. The authors write that "Our simulated c value distributions bear a moderate resemblance to our observed c value distributions". This conclusion is central to the paper's claim that "Gene regulation gravitates towards either addition or multiplication when combining the effects of two signals" (title) and that "the combined responses exhibited a range of behaviors, but clearly favored both additive and multiplicative combined transcriptional responses" (Abstract). However, the additional peak at c=1 is not obvious from the data in Figure 1E. Stronger evidence (i.e. statistical analysis of the observed distributions) would be needed to demonstrate overrepresentation of c values ~1. Alternatively, the title and Abstract could be revised to better reflect the strength of the findings.

The reviewers are absolutely right that a rigorous c-value analysis is critical for the claims we are making in the paper, and that our treatment of the c-values could have been considerably more rigorous. To make our conclusions more sound, we performed several additional analyses of our data to show that the c-value distribution has a strong overrepresentation at c=1 (Figure 1F; Figure 1—figure supplement 4). The details are in the Materials and methods, but the overall scheme was to fit the primary additive peak (c=0) to an additive model, and then subtract that distribution from the actual observed distribution. We then fit a Gaussian to the remainder, finding that the residuals contained a peak at or near c=1 for both the medium and high dose (1.28 for low-dose).

We updated our main text to describe the new analyses performed above, which can be seen below:

“At all doses of combination treatment, we observed a wide peak centered around c=0 (additive), with a hint of a secondary peak at c=1 (multiplicative), suggesting that the integration of the effects of two signals is preferentially additive or multiplicative (Figure 1F; Figure 1—figure supplement 4).[…] Overall, while there is the possibility of further peaks within our data, our data most strongly support the existence of two peaks in the c-value histogram, one corresponding most closely with an additive model, and the other with a multiplicative model.”

We thank the reviewers for pointing out this deficiency in our original manuscript, and feel that these additional analyses have greatly enhanced our manuscript.